



# Airborne observations of far-infrared upwelling radiance in the Arctic

Quentin Libois[1], Liviu Ivanescu[1,2], Jean-Pierre Blanchet[1], Hannes Schulz[3], Heiko Bozem[4], W. Richard Leaitch[5], Julia Burkart[6], Jonathan P. D. Abbatt[6], Andreas B. Herber[3], Amir A. Aliabadi[7], and Éric Girard[1]

[1]Department of Earth and Atmospheric Sciences, Université du Québec à Montréal, Montréal, Canada
[2]Centre d'applications et de recherches en télédétection (CARTEL), Université de Sherbrooke, Sherbrooke, Canada
[3]Alfred Wegener Institute, Helmholtz-Center for Polar and Marine Research, Bremerhaven, Germany
[4]Johannes Gutenberg University of Mainz, Institute for Atmospheric Physics, Mainz, Germany
[5]Environment and Climate Change Canada, Toronto, Canada
[6]Department of Chemistry, University of Toronto, Toronto, Canada
[7]Atmospheric Innovations Research (AIR) Laboratory, School of Engineering, University of Guelph, Guelph, Canada

*Correspondence to:* Quentin Libois (libois.quentin@uqam.ca)

**Abstract.** The first airborne measurements of the Far-InfraRed Radiometer (FIRR) were performed in April 2015 during the panarctic NETCARE campaign. Vertical profiles of spectral upwelling radiance in the range 8 - 50 $\mu$m were measured in clear and cloudy conditions from the surface up to 6 km. The clear-sky profiles highlight the strong dependence of radiative fluxes to the temperature inversion typical of the Arctic. Measurements acquired for total column water vapor from 1.5 to 10.5 mm

also underline the sensitivity of the far-infrared greenhouse effect to specific humidity. The cloudy cases show that optically thin ice clouds increase the cooling rate of the atmosphere by a factor up to three, making them important pieces of the Arctic energy balance. One such cloud exhibited a very complex spatial structure, characterized by large horizontal heterogeneities at the kilometre-scale. This emphasizes the difficulty to obtain representative cloud observations with airborne measurements, but also points out how challenging it is to model polar clouds radiative effects. These radiance measurements were successfully

compared to simulations, suggesting that state-of-the-art radiative transfer models are suited to study the cold and dry Arctic atmosphere. Although FIRR *in situ* performances compare well to its laboratory performances, complementary simulations show that upgrading the FIRR radiometric resolution would greatly increase its sensitivity to atmospheric and cloud properties. Improved instrument temperature stability in flight and expected technological progress should help meet this objective. The campaign overall highlights the potential for airborne far-infrared radiometry and constitutes a relevant reference for future

similar studies dedicated to the Arctic, and for the development of spaceborne instruments.

## 1 Introduction

Since the early days of weather satellites, remote sensing in the infrared (IR) has been used to study the vertical structure of the atmosphere (e.g. Conrath et al., 1970). Most instruments currently deployed, such as the Moderate Resolution Imaging Spectroradiometer (MODIS, King et al., 2003), the Atmospheric Infrared Sounder (AIRS, Aumann et al., 2003), and the





Infrared Atmospheric Sounding Interferometer (IASI, Blumstein et al., 2004), do not measure atmospheric radiation beyond approximately 15 $\mu$m, though, because sensing far-infrared radiation (F-IR, 15 $\mu$m $< \lambda <$ 100 $\mu$m) generally requires a different technology (Mlynczak et al., 2006). However, probing the atmosphere in the F-IR could provide valuable information and complement current observations. The F-IR range hosts the purely rotational bands of water vapor. It is also where the

water vapor continuum is strongest (Shine et al., 2012). As such, it is especially promising for remote sensing of water vapor in the coldest regions of the atmosphere, that is the upper troposphere and the stratosphere (Rizzi et al., 2002; Shahabadi and Huang, 2014), and the polar regions in general (Turner and Mlawer, 2010; Blanchet et al., 2011; Palchetti et al., 2015). The maximum of the Planck's function shifts towards the F-IR with decreasing temperature, so that increasingly more energy is emitted from this spectral region (Merrelli and Turner, 2012) compared to the more widely used 6.7 $\mu$m vibrational-rotational

band (Susskind et al., 2003). Hence in cold atmospheres more than half of the radiation is lost to space from the F-IR domain (Clough et al., 1992). This is one of the reasons why the Mars Climate Sounder (McCleese et al., 2007) and the Diviner Lunar Radiometer Experiment (Paige et al., 2010) measure F-IR to probe the very cold atmosphere of Mars and the moon surface, respectively. The F-IR signature of clouds also carries much information about cloud phase, optical thickness, particle size distribution and particle shape for ice clouds (Rathke, 2002; Yang, 2003; Baran, 2007). This assessed sensitivity has recently

stimulated the development of retrieval algorithms for ice cloud properties (e.g. Blanchard et al., 2009). Observing long term changes in the F-IR emission of Earth could eventually provide valuable insight into the physical processes underlying climate change (Huang et al., 2010).

    As a consequence, in the last three decades a number of scientific teams have demonstrated the need for improved observation of the Earth in the F-IR (e.g. Mlynczak et al., 2002; Harries et al., 2008). In the meantime, several F-IR spectrometers were

developed. The Atmospheric Emitted Radiance Interferometer (AERI, Knuteson et al., 2004) has been extensively used for atmospheric profiling and cloud remote sensing (Turner and Löhnert, 2014; Cox et al., 2014). The Far-InfraRed Spectroscopy of the Troposphere (FIRST, Mlynczak et al., 2006) and the Radiation Explorer in the Far-InfraRed-Prototype for Applications and Development (REFIR-PAD, Palchetti et al., 2006) were developed within the framework of the satellite projects Climate Absolute Radiance and Refractivity Observatory (CLARREO, Wielicki et al., 2013) and REFIR (Palchetti et al., 1999), respectively.

These instruments primarily aim at better constraining the radiative budget of the atmosphere, and have been operated from gondola and from the ground (Bianchini et al., 2011; Mlynczak et al., 2016). The Tropospheric Airborne Fourier Transform Spectrometer (TAFTS, Canas et al., 1997) has been used to explore the radiative properties of water vapor (Green et al., 2012; Fox et al., 2015) and to investigate the radiative properties of cirrus clouds (Cox et al., 2010). So far, all these spectrometers have been extensively used to improve the parameterization of the water vapor absorption lines and continuum in the F-IR (De-

lamere et al., 2010; Liuzzi et al., 2014), in order to refine radiative transfer codes (Mlawer et al., 2012) and climate simulations (Turner et al., 2012).

    Further understanding the radiative properties of the atmosphere in the F-IR is of uttermost importance in the Arctic because atmospheric cooling essentially occurs in this spectral range (Clough et al., 1992). Although F-IR spectrometers have been used from the ground in Alaska and Northern Canada (Mariani et al., 2012; Fox et al., 2015), we are not aware of any such airborne



measurements in the Arctic. The panarctic NETCARE (Network on Climate and Aerosols: Addressing Key Uncertainties in Remote Canadian Environments, http://www.netcare-project.ca) aircraft campaign, that took place in April 2015, attempted to fill this gap. This four-week campaign involved the two instrumented Basler BT-67 Polar 5 and Polar 6 aircraft (e.g. Ehrlich and Wendisch, 2015) and investigated the radiative properties of the atmosphere in clear and cloudy conditions. While most reported airborne F-IR observations consist of constant altitude flights, vertical profiles of spectral radiance are very instructive to understand the vertical structure of the energy budget of the atmosphere (Mlynczak et al., 2011). For this reason, most measurements taken with the Far-InfraRed Radiometer (FIRR, Libois et al., 2016) during the campaign consisted of vertical profiles of upwelling radiance from the surface up to about 6 km. The FIRR was developed as a technology demonstrator for the Thin Ice Clouds in Far-InfraRed Experiment (TICFIRE, Blanchet et al., 2011) satellite mission, whose primary focus is on the water cycle in the Arctic, and on ice clouds in particular. Like cirrus at mid-latitudes (Cox et al., 2010; Maestri et al., 2014), ice clouds encountered in the Arctic significantly affect the atmosphere radiative budget in the F-IR, especially because they can fill the whole troposphere (Grenier et al., 2009). Unlike the tropics, such ice cloud layers occur at any altitude, from the ground to the stratosphere (polar stratospheric clouds). Their impact is also very dependent on moisture (Cox et al., 2015), making the interactions between water vapor and Arctic clouds particularly complex.

In the context of TICFIRE, flying the FIRR in the Arctic served four main objectives: 1) assessing the FIRR radiometric performances in airborne conditions meant to mimic as closely as possible satellite nadir observations; 2) validating radiative transfer simulations in the F-IR for clear and cloudy Arctic conditions through radiative closure experiments; 3) verifying the spectral signature of clouds radiance *in situ*; 4) investigating the sensitivity of FIRR measurements to atmospheric characteristics and better understanding the radiative budget of the Arctic atmosphere. The FIRR measurements taken during the campaign are presented in Section 2, along with complementary observations relevant to the radiative properties of the Arctic atmosphere. Five case studies are then detailed in Section 3 and serve as a basis to assess FIRR performances in airborne conditions and explore its sensitivity to atmospheric conditions. The sensitivity to temperature, humidity and cloud properties is further investigated in Section 4 using radiative transfer simulations. The impact of an optically thin ice cloud on atmospheric cooling rates is also discussed. Based on this unique experience, recommendations are provided for future operations of instruments similar to the FIRR in such airborne campaigns.

## 2 Materials and methods

### 2.1 The NETCARE campaign

The panarctic PAMARCMIP/NETCARE campaign (hereinafter NETCARE campaign) comprises many collaborators including the following institutions: Alfred Wegener Institut (AWI), University of Toronto, Environment and Climate Change Canada (ECCC), and more parties listed under the website. The campaign started in Longyearbyen (Spitzbergen) on 5 April with one week delay due to harsh weather conditions, with a single flight dedicated to calibration. Then the aircraft flew across Station North (Greenland) and operated around Alert, Nunavut (Canada) from 7 to 9 April. Afterwards, they moved to Eureka,



Nunavut (Canada) and stayed there until 17 April. They continued to Inuvik, Northwest Territories (Canada), where Polar 6 operated until 21 April, while Polar 5 headed towards Barrow (Alaska). No flights were performed with Polar 6 from 14 to 19 April, due to cloudy conditions at Eureka and technical problems with the aircraft. The two aircraft had different scientific objectives, with Polar 5 mostly dedicated to sea ice studies and Polar 6 to aerosol and cloud studies. In the following, only

Polar 6 operations are detailed, which consist of 10 scientific flights, amounting to 50 hours of campaign flight time.

The NETCARE campaign aimed at better understanding aerosol transport into the Arctic in the early spring, and its influence on ice cloud formation. Many instruments were installed aboard Polar 6, including basic meteorology and radiation sensors, cloud microphysics instrumentation, particle counters, trace gas monitors and instruments for monitoring aerosol composition (e.g. Leaitch et al., 2016). Each flight was planned based on forecasts of clouds and transported pollution as well as the location

of the A-Train satellite constellation (Stephens et al., 2002). The atmosphere was generally probed vertically from the surface (∼50 m) to approximately 6000 m (or the other way round) in about 50 min. To this end, the aircraft followed quasi-spirals of diameter 10 km.

### 2.2 *In situ* observations

#### 2.2.1 The Far-InfraRed Radiometer (FIRR)

The FIRR (Libois et al., 2016) uses a filter wheel to measure atmospheric radiation in 9 spectral bands ranging from 8 to 50 $\mu$m (Table 1). The FIRR sensor is a 2-D array of uncooled microbolometers coated with gold black (Ngo Phong et al., 2015), and radiometric calibration is achieved with two reference blackbodies (BB) at different temperatures. The latter consist of cavities whose temperature and emissivity are well known, so that the radiance they emit is accurately estimated. During the NETCARE campaign, the FIRR was onboard Polar 6 and measured upwelling radiance directly through a 56 cm-long

vertical chimney (Fig. 1a). At the bottom of the chimney, a rolling door opened during the flight (Fig. 1b), but remained closed otherwise to prevent dust or blowing snow from entering the instrument. Although the FIRR has a nominal field of view of 6° corresponding to a 20 pixels diameter area, here only a 15 pixel diameter area is used to avoid vignetting on the edges. This corresponds to a field of view of 4.5°, which translates into a footprint of 7.8 m at a 100 m distance, and 470 m at 6000 m. Since the temperature aboard the unpressurized cabin quickly varied between approximately 0 and 15°C, the ambient

blackbody (ABB) was maintained at 15°C, while the hot blackbody (HBB) was set to 45 or 50°C. One FIRR measurement sequence lasts 3 min 30 s and consists of two calibration sequences (one on the ABB and one on the HBB) followed by 3 scene sequences, each sequence corresponding to one complete rotation of the filter wheel that measures all 9 filters. For each spectral band, 100 frames are acquired at 120 Hz and then averaged to provide a single 2-D image. One measurement thus corresponds to a 0.8 s long acquisition and no supplementary temporal average is performed, highlighting the potential

for fast scanning compared to interferometers that usually require averaging over several spectra to achieve comparably high performances (e.g. Mlynczak et al., 2006). Such acquisition rate is essential when looking at heterogeneous or quickly moving targets, as is the case from an aircraft or satellite view. It is the main advantage of trading spectral resolution for higher signal levels. In this study, the FIRR is not used as an imager, hence the data are also spatially averaged over the selected area of 193




**Table 1.** FIRR spectral bands.

| Band number | Spectral range (μm) | Band number | Spectral range (μm) |
|---|---|---|---|
| 1 | $7.9 - 9.5$ | 6 | $17.25 - 19.75$ |
| 2 | $10 - 12$ | 7 | $20.5 - 22.5$ |
| 3 | $12 - 14$ | 8 | $22.5 - 27.5$ |
| 4 | $17 - 18.5$ | 9 | $30 - 50$ |
| 5 | $18.5 - 20.5$ | | |

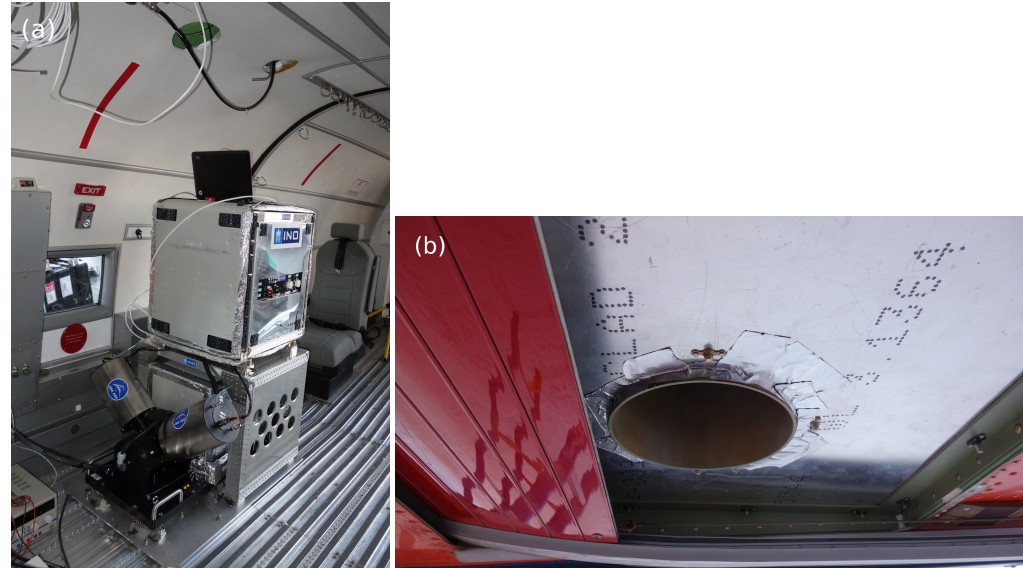

**Figure 1.** (a) The FIRR setup aboard Polar 6. The optomechanical device is on the floor while the electronic components are above in the insulated rack. (b) The rolling door at the bottom of the chimney through which FIRR takes measurements. The door is shown in optimal position for instrument stability, but nominal position is completely on the left. Flight direction is towards the left.

pixels. In this configuration, the radiometric resolution of the FIRR in laboratory conditions is about $0.015 \text{ W m}^{-2} \text{ sr}^{-1}$, which corresponds to noise equivalent temperature differences of $0.1 - 0.35$ K for the range of temperatures investigated in this study. Such performances compare well with similar airborne spectroradiometers (e.g. Emery et al., 2014) and satellite sensors (e.g. MODIS).

5      A critical issue during the campaign was the temperature stability of the instrument in operation. Indeed, the first flights were characterized by excessively noisy measurements, especially in the $30 - 50$ $\mu$m channel. This noise was due to excessive air circulation within the chimney, cooling down very quickly the calibration enclosure and the filters. In particular, the metallic mesh filter $30 - 50$ $\mu$m has a very low thermal capacity and its temperature significantly changed in less than 1 s, making the acquired data unusable. A float-zone silicone window was available that could be placed at the entrance of the instrument,

10    but we decided not to use it since its limited transmittance of $30\%$ in the F-IR drastically reduced signal level. This issue was





fixed on 13 April by partially closing the rolling door in flight to prevent cold air flow from entering the inlet chimney, without impacting the field of view. For previous flights, the calibration procedure detailed in Libois et al. (2016) was refined to better account for quick temperature variations. It takes advantage of non illuminated pixels of the detector to remove the background signal, and ensured good quality data for all bands except the $30 - 50 \ \mu$m.

### 2.2.2 Other measurements

Polar 6 was equipped with a large set of sensors and instruments but only those relevant for the present study are mentioned below. Air temperature was recorded with an accuracy of 0.3 K by an AIMMS-20 manufactured by Aventech Research Inc. (Aliabadi et al., 2016). Trace gas $H_2O$ measurement was based on infrared absorption using a LI-7200 enclosed $CO_2/H_2O$ Analyzer from LI-COR Biosciences GmbH. In-situ calibrations during the flights were performed on a regular time interval of 15 to 30 min using a calibration gas with a known $H_2O$ concentration close to zero. The uncertainty for the measurement of $H_2O$ is 39.1 ppmv or 2.5 %, whichever is greater. Broadband longwave (LW) radiation was measured with Kipp & Zonen CGR-4 pyrgeometers installed below and above the aircraft (Ehrlich and Wendisch, 2015). These sensors have uncertainties of a few W m$^{-2}$. Nadir brightness temperature in the range $9.6 - 11.5 \ \mu$m was measured by a Heitronics KT19.85 II with a field of view of 2° and an accuracy of 0.5 K. A number of probes also provided qualitative information about the presence of cloud particles. Total and liquid water content were measured with a Nevzorov probe (Korolev et al., 1998). An FSSP-300 particle probe was used to measure particle size distributions from 0.3 to 20 $\mu$m from which cloud presence can be deduced (e.g. Ström et al., 2003). A PMS 2D-C imaging probe was used to detect larger particles. However the images were obscured due to a problem with the true air speed used in the image re-construction, preventing accurate retrieval of particle size distribution. A sun-photometer specially designed for Polar 6 (SPTA model by Dr. Schulz & Partner GmbH) was mounted on top of the aircraft and continuously tracked direct solar radiation in 10 spectral bands in the range $360 - 1060$ nm. From these spectral measurements, the atmospheric optical depth was deduced and further processed with the SDA method (O'Neill et al., 2003) to retrieve the contributions of the fine (aerosols) and coarse (mainly cloud and precipitation) mode components. In addition to these particle measurements, black carbon concentration was estimated to give an indication on the level of pollution of the investigated air masses. To this end, ambient air was sampled with an inlet mounted above the cockpit of Polar 6, and a Single Particle Soot Photometer (SP2 by Droplet Measurement Technologies, Boulder, Colorado) was used to evaluate the mass of individual refractive black carbon particles per volume of air (Schwarz et al., 2006), from which the mass for particles within the size range $75 - 700$ nm was deduced. High resolution nadir pictures taken at 15 s intervals also provided valuable information about the surface and the presence of clouds.

### 2.3 Selected flights

For the present study, 5 vertical profiles taken during 5 different flights were selected. These flights, whose trajectories are shown in Fig. 2, were performed near Alert (82.5° N, 62.3° W), Eureka (80° N, 86.1° W) and Inuvik (68.3° N, 133.7° W) on 7, 11, 13, 20 and 21 April. All profiles were measured above snow-covered sea ice, which ensured that the surface was homogeneous contrary to flights performed above patches of snow and tundra or over areas of mixed sea ice and open water.





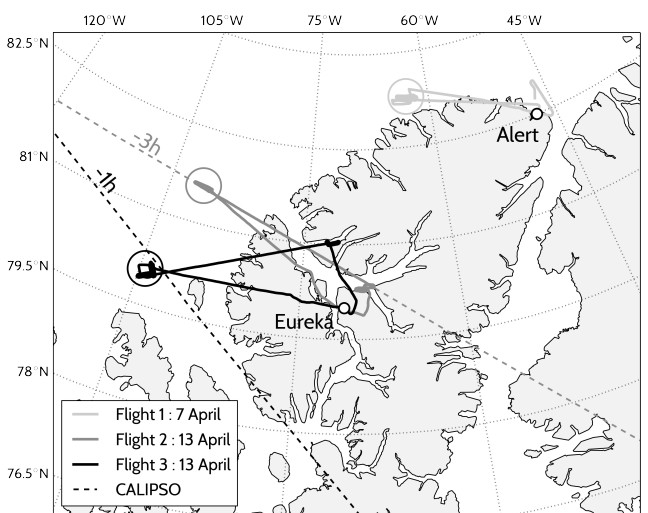
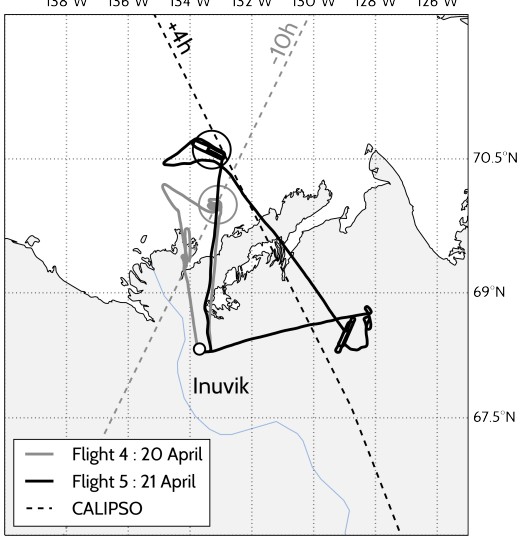

**Figure 2.** Selected flight trajectories around (left) Eureka and (right) Inuvik. The circles indicate where the detailed vertical profiles were performed. CALIPSO tracks are also shown and hours indicate how much earlier (-) or later (+) the satellite flew over.

All the investigated flights except 7 April were taken close to a track of the Cloud-Aerosol Lidar and Infrared Pathfinder Satellite Observations satellite (CALIPSO , Winker et al., 2003). The 5 profiles were acquired in distinct atmospheric conditions, thus providing a valuable overview of Arctic conditions in early spring. April 7 to 13 flights corresponded to typical conditions of the high Arctic cold season, with low temperatures and a pronounced inversion, while the conditions near Inuvik were more

representative of subarctic spring, with near-melting temperatures at the surface and denser clouds typically found in the mid-latitudes. Some ice clouds were encountered on 7 April flight, but the more typical polar optically thin ice cloud was probed on 13 April near Eureka. The three other flights exhibited clear sky conditions below the aircraft.

## 2.4 Radiative transfer simulations

One objective of the study was to perform radiative closure experiments by comparing FIRR measurements with radiative

transfer simulations based on thermodynamical and microphysical profiles recorded by the instruments aboard Polar 6. Here we used MODTRAN v.5.4 (Berk et al., 2005) to simulate upwelling radiance at flight level. MODTRAN uses absorption lines from HITRAN2013 and the MT-CKD 2.5 parameterization of the water vapor continuum (Clough et al., 2005) that proved reliable in the Arctic (Fox et al., 2015). The spectral surface emissivity of snow was taken from Feldman et al. (2014). Aerosols are approximated to the standard rural profile with a visibility of 23 km which is consistent with the presence of Arctic

haze during the campaign. Multiple scattering is computed with DISORT (Stamnes et al., 1988) using 16 streams, and the band model is at 1 cm$^{-1}$ spectral resolution. The model atmosphere has 75 levels from the surface to 30 km, with a resolution of



0.1 km near the surface stretching to 0.7 km at the top. In addition to radiances, MODTRAN was used to compute atmospheric cooling rates (e.g. Clough et al., 1992) and Jacobians through finite differences (Garand et al., 2001).

Temperature and humidity profiles were interpolated from the *in-situ* measurements up to the maximum flying altitude. Above, they were taken from the closest ERA-Interim reanalysis (Dee et al., 2011), the latter being offset to ensure vertical
continuity. Ozone profiles for the whole column were also taken from ERA-Interim. Snow surface temperature was obtained from the KT19 observations assuming a surface emissivity of 0.995. All simulated clouds in this study are ice clouds defined by their optical thickness $\tau$ and effective particle diameter $d_{\mathrm{eff}}$. Their single scattering properties are calculated after the parameterization of Yang et al. (2005) for cirrus clouds. Cloud geometrical characteristics were deduced from the combination of *in situ* observations. Optical thickness and effective cloud particle diameter were not directly measured. For 7 April, both
quantities were tuned to minimize the deviation from measurements. For 13 April, the effective particle diameter was taken from DARDAR satellite product (Delanoë and Hogan, 2010) and simulations were performed for various optical depths.

## 3 Results

In this section, the FIRR radiometric performances are first analyzed based on experiments performed on the ground and during one flight. The five case studies are then analyzed in detail and the vertical profiles of radiance acquired in clear sky and cloudy
conditions are compared to radiative transfer simulations.

### 3.1 FIRR radiometric performances in airborne configuration

The FIRR performances were investigated based on laboratory and ground-based experiments by Libois et al. (2016), who estimated a radiometric resolution around 0.015 W m$^{-2}$ sr$^{-1}$. In airborne configuration, the environmental conditions were more demanding due to cold ambient temperature and quick background temperature variations. The FIRR performances for
this specific setup are thus estimated from two experiments for which the environmental conditions were similar to nominal airborne operation, except the scene was more constant than in operation. Firstly, the brightness temperature of the snow surface below the aircraft was measured on Eureka runway on 12 April, while Polar 6 was parked without the propellers running. The ambient temperature was around -32°C, the ABB was at -9.5°C and the HBB at 20°C. Secondly, measurements taken on the closed rolling door just before landing on 11 April were analyzed. For this case, the ABB was at 15°C and the HBB at 45°C.
Although the BB temperatures were different than nominal values, this has no impact on the results since the instrument's response is highly linear and because the temperature difference between the HBB and ABB remained nearly constant.

The experiment on snow consisted of 10 consecutive measurement sequences covering 30 min, so that 30 radiances were recorded for each spectral band. Those were first detrended to remove the effect of snow temperature variations over the period, and the standard deviation of the residual was then computed. The latter does not exceed 0.012 W m$^{-2}$ sr$^{-1}$. The experiment
performed on the rolling door consisted of 5 consecutive sequences, and the standard deviation of the signal was larger, reaching 0.021 W m$^{-2}$ sr$^{-1}$. Figure 3a shows the corresponding brightness temperatures for both experiments, highlighting a temperature resolution around 0.1 K above snow and 0.2 K above the rolling door. Although the environmental conditions



are slightly different in flight, these results provide a valuable reference and show that the installation of the instrument in the aircraft did not affect its performances.

To further investigate the reduced resolution observed in flight, Fig. 3b shows the sequence of brightness temperatures recorded on the rolling door. A recurrent pattern is observed within a sequence of 3 consecutive measurements, with the

first temperature generally larger than the following ones. We interpret this as the signature of fast and complex temperature variations of the skin temperature of the filter, that cannot be removed through the calibration procedure. We attempted to use the numerous temperature sensors embedded in the calibration enclosure and in the filter wheel to reconstruct the filters actual temperature, but this proved unsuccessful. Without any indication of whether any of the 3 consecutive points is the best, we simply conclude that this thermal instability results in an additive noise of approximate amplitude 0.2 K in worst conditions.

This leaves room for future improvement of the instrument. The operational resolution of the FIRR nevertheless remains well below 0.5 K, which is still satisfactory and comparable to temperature measurements performed aboard Polar 6. This issue had not been noticed by Libois et al. (2016), most likely because in their study ambient temperature was closer to the internal temperature of the FIRR, limiting the range of filter temperature variations.

### 3.2   Clear sky cases

The profiles on 11, 20 and 21 April were all taken in clear sky conditions, but the total columns of water vapor were very different. These flights are specifically used to investigate the impact of temperature and humidity variations on the measured profiles of spectral radiances.

#### 3.2.1   11 April

The ascent started at 19:02 UTC and at 19:52 UTC Polar 6 reached the maximum altitude of 5.56 km, where it stayed for

4 min. On its way up it also levelled at 2.75 km for 7 min. The surface temperature retrieved from the KT19 was -32.6°C and a maximum of -24°C was observed from 1 to 2 km (Fig. 4a). The whole atmosphere was undersaturated with respect to ice, except near the surface. The total column water vapor was 1.5 mm, with 1.4 mm below 5.56 km. No clouds were observed and the CALIPSO profile taken 3 hours earlier suggests that no clouds were present above either. FIRR brightness temperature profiles show interesting features (Fig. 4b), with the temperature inversion more obvious for the longer wavelengths

for which the atmosphere is more opaque. The $17 - 18.5$ $\mu$m and $18.5 - 20.5$ $\mu$m profiles are very similar, implying relative redundancy between these two channels. As expected, the brightness temperature in the highly transparent atmospheric window ($10 - 12$ $\mu$m) is essentially constant with height since it is insensitive to the properties of the atmosphere. The slight increase of 0.5 K from the surface to the top is also observed in KT19 records and is probably the signature of surface temperature variations. The very distinct behaviors of window and F-IR channels still result in nearly similar brightness temperatures at the

top of the profile. This feature, typical of the Arctic, highlights the complexity of probing from space an atmosphere with a strong temperature inversion. The peaks in the shorter wavelengths channels around 4 km were found to visually correspond to variations of sea ice characteristics. They could be due to thinner and warmer sea ice or finer snow with higher emissivity (Chen et al., 2014). Since all individual measurements were used, the vertical resolution is close to 200 m. However, the instability




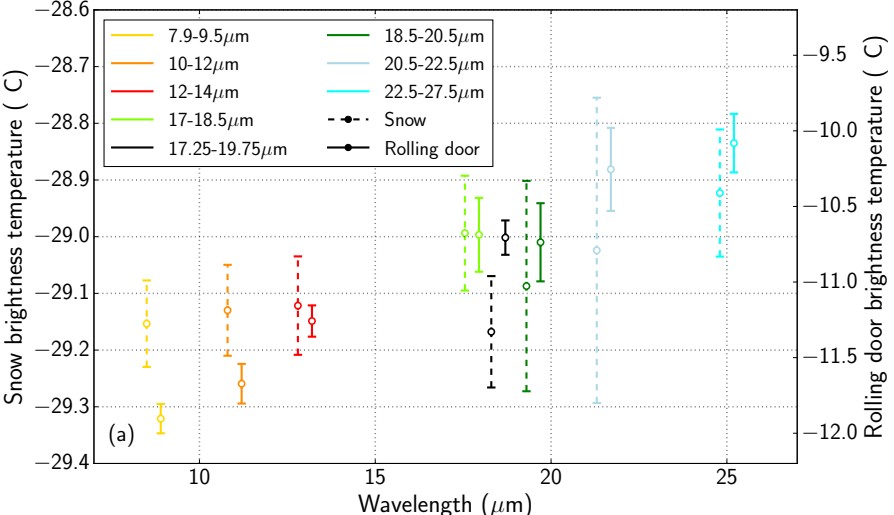

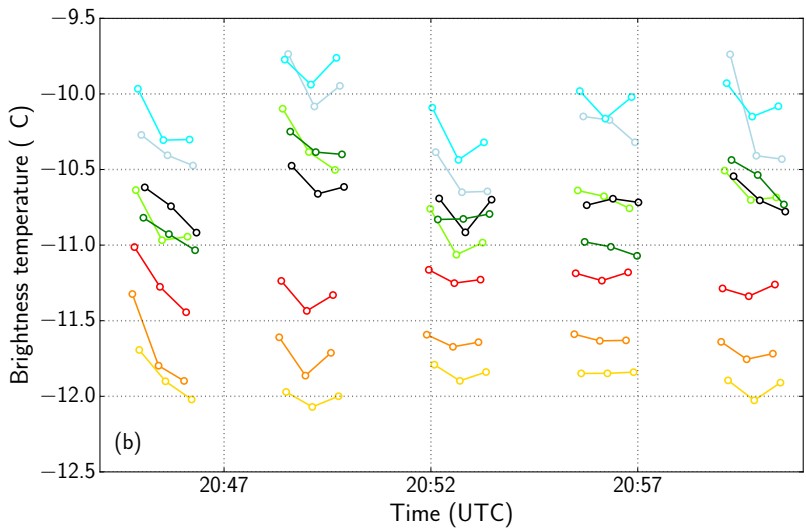

**Figure 3.** (a) Mean and standard deviations (error bars) of the detrended brightness temperatures along 10 sequences (i.e. 30 consecutive measurements) for measurements taken on snow on 12 April ($15:10-15:42$ UTC) and along 5 sequences on the rolling door on 11 April 11 ($21:45-22:00$ UTC). For 12 April, $T_{HBB} = 20°C$ and $T_{ABB} = -9.5°C$. For 11 April, $T_{HBB} = 45°C$ and $T_{ABB} = 15°C$. (b) Temporal evolution of brightness temperature for the 5 sequences acquired on the closed rolling door on 11 April. The $30-50$ $\mu$m band is not shown because it suffered from the temperature stability problem mentioned in Section 2.2.1.





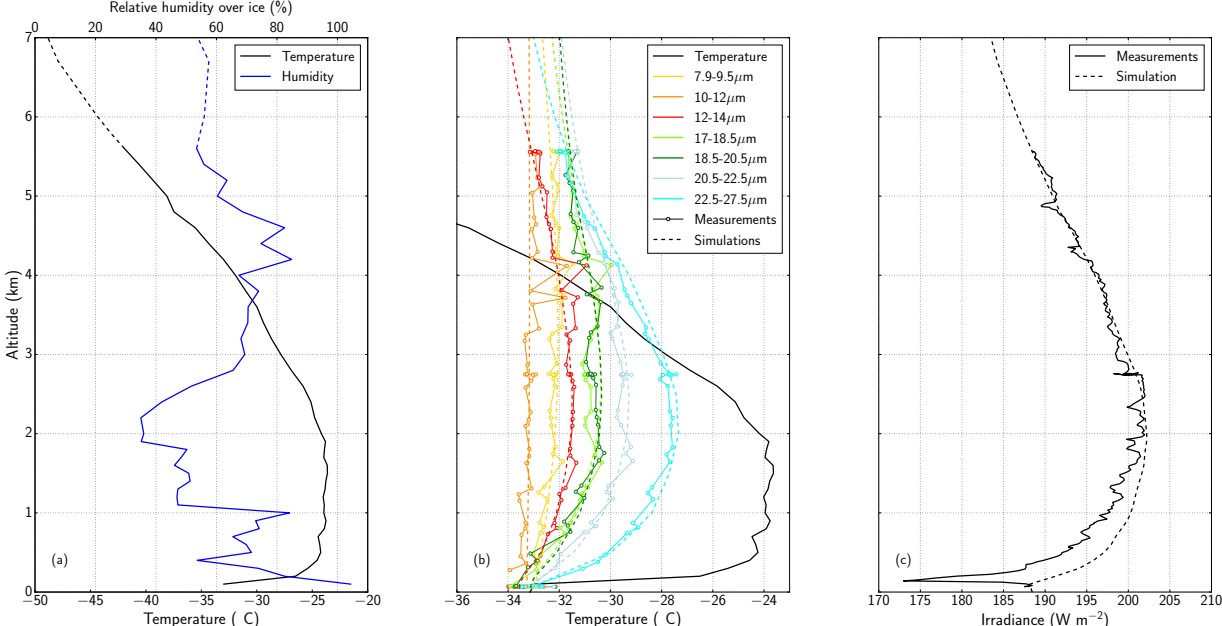

**Figure 4.** Vertical profiles of (a) temperature and relative humidity, (b) FIRR brightness temperatures and (c) upwelling broadband LW irradiance for 11 April flight. The simulated FIRR brightness temperatures and LW irradiance are also shown. The $17.25 - 19.75$ $\mu$m band is not shown because it overlaps with others. The dashed lines in panel (a) correspond to the ERA-Interim profiles used for the simulations above maximum flying altitude.

along 3 measurements is noticeable, for e.g. the $18.5 - 20.5$ $\mu$m channel below 2 km. Besides this instrumental noise, part of the observed signal variation might be due to horizontal inhomogeneity, especially when the aircraft roll reaches up to 20° in turns.

The vertical profile of upwelling broadband LW radiation also highlights the temperature inversion, with a maximum around
5   2 km, similar to the F-IR channels of the FIRR (Fig. 4c). LW fluxes have been simulated with MODTRAN and are also shown. The simulated and measured profiles are in very close agreement above 1 km, with a root mean square deviation (RMSD) of 0.6 W m$^{-2}$. Such a value is consistent with the accuracy provided by the manufacturer and the absolute uncertainty of 2 W m$^{-2}$ suggested by Marty (2003) for such sensors. This is very satisfactory for a sensor sensitive only up to 42 $\mu$m while a significant part of the energy lies beyond, and considering that the calibration was done above 2°C. This agreement gives
10  high confidence in the atmospheric profile measurements, but also in the aerosols modelled in MODTRAN, because errors in aerosol profiles could result in discrepancies of several W m$^{-2}$ (Sauvage et al., 1999). Regarding the upper extrapolated part of the atmosphere, comparison of measured and simulated downwelling LW fluxes (not shown) are also in reasonably good agreement, which gives confidence in the ERA-Interim fields. Close to the surface, measurements show an unexpected





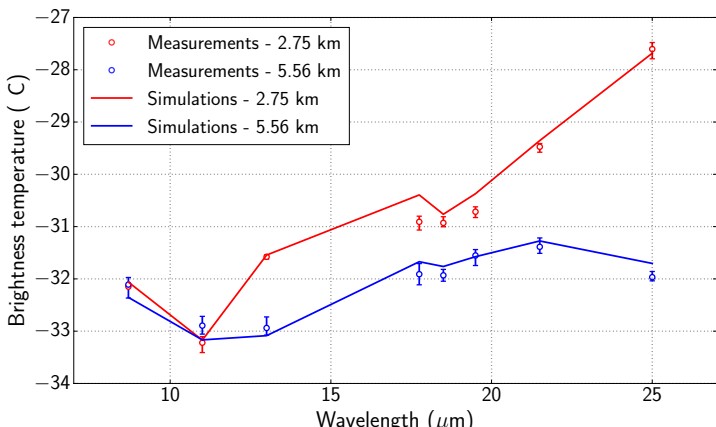

**Figure 5.** Measured and simulated spectral brightness temperatures at the two altitudes where Polar 6 levelled during 11 April flight. At both levels 4 consecutive measurements were taken. Their means and ranges are indicated by the circles and error bars, respectively.

behavior. Although the origin of this feature is not understood, we believe this is an instrumental artifact resulting from the strong temperature gradient near the surface, and the sensor not being at thermal equilibrium (Ehrlich and Wendisch, 2015). This hypothesis is supported by the fact that data taken on the way down just before starting the ascent do not show the same feature.

MODTRAN was also used to simulate FIRR brightness temperatures (Fig. 4b). The measured profiles for all channels are well simulated, with a mean bias and RMSD below 0.2 K. The agreement in the window bands confirms that no clouds were present below the aircraft. F-IR simulations provide strong validation of the radiative transfer model. The spectral brightness temperatures are compared at the two altitudes where multiple measurements were taken. Figure 5 shows the average measured brightness temperatures at 2.75 and 5.56 km, and the corresponding simulations. The spectral RMSD is below 0.15 K at both

altitudes, which is very satisfying, given that MODTRAN user's manual suggests that the model accuracy is 1 K. The variability of the measurements at each step is below 0.4 K which is consistent with the results of Fig. 3b.

      Overall, the simulations reproduce well the observations, which validates to some extent the radiative transfer code configuration and the implemented snow emissivity. However, such measurements can hardly be used for model improvement. As pointed out by Mlynczak et al. (2016), the inherent uncertainties related to the atmospheric measurements and radiative

transfer parameterization likely exceed the FIRR measurements uncertainties. Agreement is thus satisfactory and encouraging for the performances of the instrument, but does not give further indications about the quality of the model inputs and parameterizations.





### 3.2.2 20 and 21 April

Both flights took place in the vicinity of Inuvik and showed relatively warm conditions and above freezing temperatures at the inversion level (Fig. 6a and c). The cloud probes suggested that no clouds were present, which is consistent with the relative humidity profiles. For 20 April flight, a moist layer typical of long range transport was found, that peaked near 2.5 km at about 85% humidity with respect to water. Above 3.5 km, this layer was topped with drier air associated with weak air subsidence. Above 3.8 km, the air was very whitish, and the FSSP-300 and sun-photometer indicate increased level of aerosols. Likewise, SP2 measurements showed increasing amounts of black carbon with altitude, exceeding 0.1 $\mu$g m$^{-3}$, which is indicative of a polluted air mass. Similar conditions were encountered on 21 April, except that the polluted layer was located above 2.6 km, which again coincided with a drop of relative humidity. Sun-photometer data suggest the presence of high altitude clouds with optical depth around 0.2, but characterized by large variability. Those clouds were not accounted for in the simulations.

The vertical profiles of brightness temperatures are similar for both flights (Figs. 6b and d). Again, the window channels show very weak variations, which is characteristic of clear sky conditions. On the contrary, F-IR channels are characterized by rapid variations near the surface and a larger lapse rate at higher altitude compared to the 11 April flight. These features are due to a sharper temperature inversion and a reduced transparency of the atmosphere (the column water vapor below 5.4 km are 10.3 mm and 10.5 mm, respectively). The peak observed at 3.8 km on 21 April corresponds to measurements over open water, as shown by a picture taken concomitantly (Fig. 7). More generally, since Polar 6 approximately flew at 75 m s$^{-1}$, a single measurement of 0.8 s spanned 60 m at the surface, which could generate noise if the surface was not homogeneous at this scale. This was the case at the interface between the sea ice and open water.

The simulated brightness temperatures in the atmospheric window are in good agreement with observations, but deviations exceeding measurement uncertainties are found in the F-IR channels for the upper part of the profile. The largest discrepancies are obtained in the $30 - 50$ $\mu$m band, with measurements being approximately 1.5 K warmer than the simulations. In fact, the air transmittance in this channel is so low that a significant part of the signal comes from the 56 cm-long chimney just below the instrument, rather than from the atmosphere. This artifact was noticed by Mlynczak et al. (2016). Using their correction (eq. 1), we find that air at -5°C and 50 % relative humidity in the chimney can increase the apparent brightness temperature at 5 km altitude by 1.5 K in the $30 - 50$ $\mu$m band, while the deviation does not exceed 0.3 K for the other channels. For this reason, the data in the $30 - 50$ $\mu$m band are not reliable. They are of little interest because at low altitude this band essentially probes local temperature. Consequently they are not shown in the rest of the paper. The consistent positive bias of the simulations in the other F-IR channels is more puzzling, especially because it is observed in both flights. Several factors could explain such discrepancies. Inaccuracies in the water vapor continuum are ruled out because recent studies have shown uncertainties below 10% (Liuzzi et al., 2014; Fox et al., 2015), largely insufficient to explain such differences. Errors in water vapor measurements are also unlikely, because independent measurements taken by distinct instruments aboard Polar 6 show differences less than 20%, while only an increase larger than 50% could explain the observed differences. Adding an optically thin cloud between 6 and 9 km altitude did not improve the simulations either. Given the verified accuracy of the FIRR, we hypothesize that the differences are the consequence of the observed haze layer. This is in line with with the significant radiative signature in the





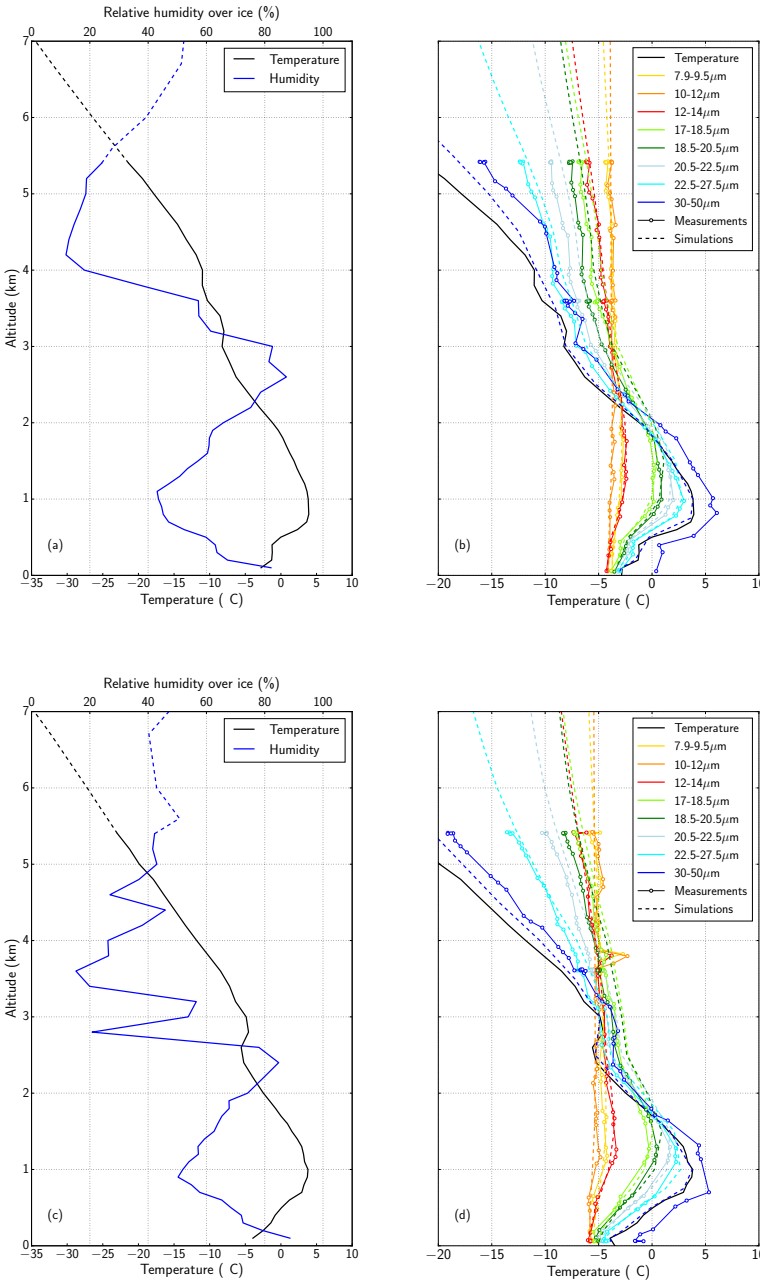

**Figure 6.** Vertical profiles of temperature and relative humidity for (a) 20 April and (c) 21 April flights. Measured and simulated FIRR brightness temperatures for (b) 20 April and (d) 21 April flights. The dashed lines in panels (a) and (c) correspond to the ERA-Interim profiles used for the simulations above maximum flying altitude.





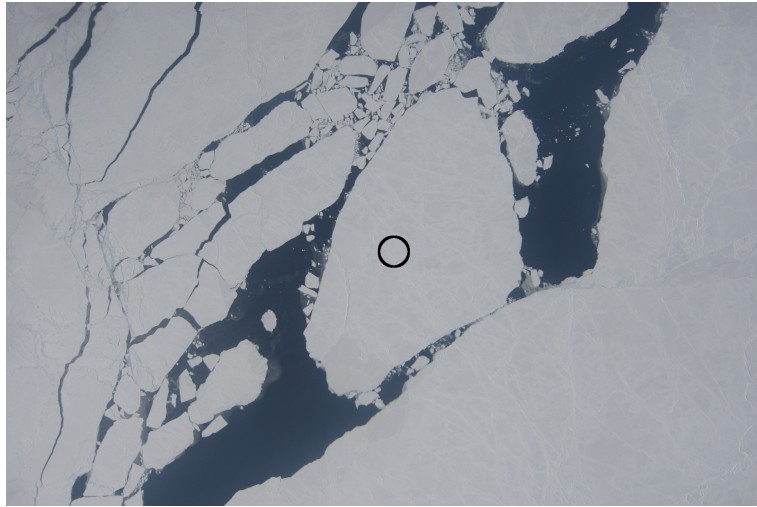

**Figure 7.** Downward picture of the surface taken on 21 April at 17:35 UTC and 3.8 km altitude. The 300 m diameter circle depicts the FIRR footprint at the surface.

IR shown by Ritter et al. (2005) for similar aerosol optical depths as those experienced in these two flights. The fact that the window channels are not impacted remains questioning, though. This might be due to the specific nature of the wet aerosols forming the haze layer, which should have a signature similar to water vapor in the F-IR. This question is let to future work, where hyperspectral measurements would certainly help investigating the detailed response. It should nevertheless be borne

in mind that in these particular cases the greenhouse effect is underestimated in MODTRAN simulations, which can lead to significant deviations on the atmospheric and surface energy budgets.

### 3.3 Cloudy cases

Flights performed on 7 and 13 April are used to assess the radiative impact of optically thin ice clouds in the F-IR. They also highlight the difficulty to compare *in situ* observations to radiative transfer simulations due to high variability of the cloud

microphysics.

#### 3.3.1 7 April

During this flight west of Alert, singular atmospheric conditions were encountered. Near the surface, a saturated layer was found up to 1.1 km where a cloud was present, as detected by the Nevzorov and 2D-C probes. Another cloud was found above 4 km, that extended up to the maximum flying altitude of 6 km. In between, the atmosphere was very dry. The temperature

profile had a complex signature near the surface, where a double temperature inversion was observed (Fig, 8), probably due to radiative cooling at top of the near-surface cloud. Observed FIRR brightness temperatures are consistent with the atmospheric profile. In the clear sky region, the profiles are similar to that of 11 April. In clouds, brightness temperature varies more rapidly





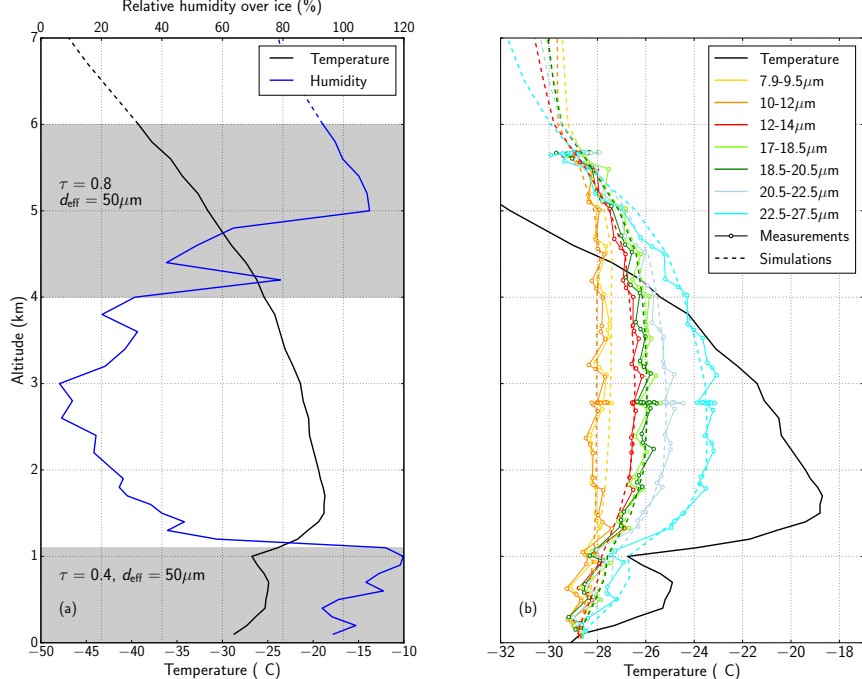

**Figure 8.** Vertical profiles of (a) temperature and relative humidity, and (b) measured and simulated FIRR brightness temperatures for 7 April flight. Shaded areas in panel (a) indicate the presence of clouds. The optical thickness and effective particle diameter used for the simulations are also indicated. The dashed lines correspond to the ERA-Interim profiles used for the simulations above maximum flying altitude.

with altitude, as a consequence of increased absorption and scattering in all channels. Consequently, all brightness temperatures samples at 5.7 km are contained in a narrow 1.5 K range.

Since CALIPSO does not cover such high latitudes, we do not have supplementary information regarding the clouds properties. The profile of relative humidity suggests that the cloud was initiated above 5 km in saturated air with respect to ice, 5   and below ice particles were precipitating, without saturating the air. For the MODTRAN simulations, the effective particle diameter was set to 75 $\mu$m, consistently with relatively large particles seen by the 2D-C probe, but missed by the FSSP-300. We then tuned the optical depth to 0.5 for the near-surface cloud layer and 1.0 for the upper layer cloud. This set of cloud properties produces brightness temperatures profiles in agreement with the measurements. The brightness temperature difference between $7.9-9.5$ $\mu$m and $10-12$ $\mu$m channels is larger in the model than in the observations yet, which suggests an imperfect 10   definition of aerosol and haze profiles.





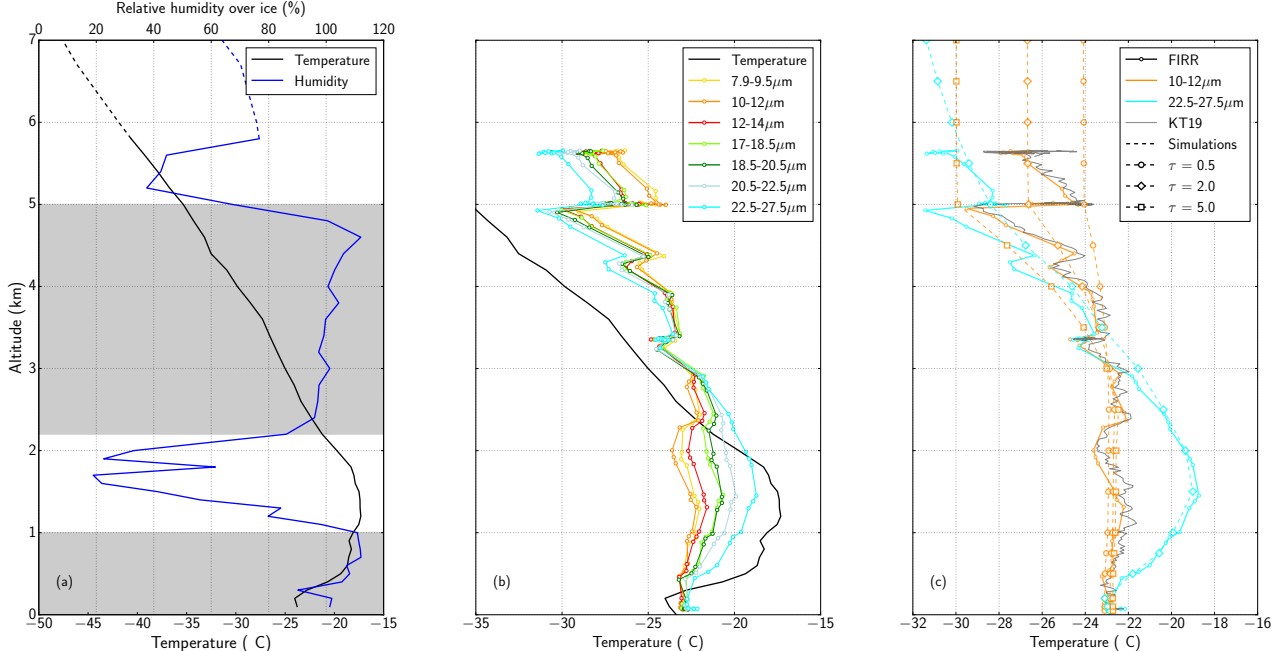

**Figure 9.** Vertical profiles of (a) temperature and relative humidity, and (b) FIRR brightness temperatures for 13 April flight. In panel (a), the shaded areas indicate the presence of clouds and the dashed lines correspond to the ERA-Interim profiles used for the simulations above maximum flying altitude.. Panel (c) shows measured and simulated brightness temperatures for two FIRR bands and various optical depths of the upper cloud. KT19 temperatures are shown as well for comparison to FIRR 10 - 12 $\mu$m channel.

### 3.3.2    13 April

The best case of optically thin ice cloud was observed during 13 April flight. A vertical profile was taken during the descent between 18:15 and 19:12 UTC. The temperature profile was fairly typical of Arctic winter conditions, with an inversion at 1.3 km and surface temperature around -25°C (Fig. 9a). A tenuous cloud layer was found below 1 km and a much thicker cloud

5   was observed between 2.2 and 5 km according to the combination of 2D-C and FSSP-300 probes. These two instruments, along with the relative humidity profile, suggest that ice particles formed above 3 km but large precipitating crystals were observed down to 2.2 km. This cloud is similar to a TIC-2B type from the classification of Grenier et al. (2009). The FIRR brightness temperatures are characterized by high vertical variability, especially above 3 km (Fig.9b). This variability is identical for all bands, suggesting that it is due to actual scene variations. The excellent match between KT19 measurements and the $10-12\,\mu$m

10  channel confirms that observed variations are not instrumental artifacts (Fig. 9c). Instead, they are attributed to cloud horizontal variability. This hypothesis is supported by the sun-photometer data that show highly varying optical depth above the aircraft as well.





Since the aircraft is flying in quasi-spirals of 10 km diameter, any cloud variability below this scale results in signal variability on the vertical profile. Downlooking pictures taken on Polar 6 show that above 3 km, surface features were intermittently visible, meaning that cloud optical depth varied substantially along the flight path. Attempting to reproduce the measured brightness temperature profiles with a 1-D model was impractical. Instead, several MODTRAN simulations were performed for various optical depths. For these simulations, effective particle diameter was set to 120 $\mu$m, consistently with DARDAR product corresponding to a CALIPSO overpass at 16:10 UTC. The near-surface cloud optical depth was set to 0.07, while the upper cloud optical depth $\tau$ was varied from 0.5 to 5 in the calculations. Figure 9c shows that the range $0.5 - 5$ reproduces quite well the observed variability of brightness temperature. We infer that at small scale cloud variability is extremely high, which is unexpected from satellite data on the large scale for this type of cloud (Grenier et al., 2009). To further investigate the spatial variability, MODIS cloud products (Platnick et al., 2003) at 18:09 UTC were analyzed. In particular, the cloud optical depth and cloud top altitude, shown in Fig. 10, are very instructive. At the scale of Polar 6 spiral, the cloud optical depth is indeed highly variable, ranging from nearly clear sky to values exceeding 5. The cloud top altitude also shows that the probed cloud with top at 5 km was very localized in the most SE section of the spiral flight. Interestingly, these spatial features are consistent with FIRR observations. In fact, the difference between the temperature measured by the $10 - 12$ $\mu$m channel and the simulation with $\tau = 2$ (indicated by the color of the trajectory in Fig. 10) is minimum in the area corresponding to the high altitude cloud. It is higher elsewhere, meaning that FIRR senses warmer temperatures corresponding to either a thinner or lower cloud. Observed FIRR spatial variability is thus consistent with the presence of a cloud of optical depth around 4 in the SE bound of the trajectory. Elsewhere on the trajectory the atmosphere ranges from clear to near-surface clouds. The latter also seem to be variable, resulting in slight variations of brightness temperature in the window channels near the surface. This case illustrates the complexity of atmospheric radiative transfer in heterogeneous conditions. It also shows that the FIRR is responding consistently with variations in clouds conditions from a nadir view similar to a satellite view.

## 4 Discussion

The five case studies investigated in the previous section provided a valuable insight on FIRR performances from an airborne nadir configuration, and on the F-IR characteristics of the Arctic atmosphere in clear and cloudy conditions. To further explore the dependence of FIRR measurements on atmospheric profiles, a series of radiative transfer simulations are performed. The radiative impact of an ice cloud is then investigated in terms of atmospheric cooling rates. The results are discussed in the framework of TICFIRE, with the intent to improve the data quality in future similar airborne campaigns.

### 4.1 Sensitivity to temperature, humidity and cloud properties

In order to extend the interpretation of the data acquired during the NETCARE campaign, the Jacobians of the top of atmosphere (TOA) brightness temperature with respect to temperature and humidity were computed for 11 April simulations (Fig. 11). The Jacobian at a given atmospheric level is the difference in simulated TOA brightness temperature resulting from an increase of 1 K (1% specific humidity) of the temperature (humidity) at this level. The temperature Jacobians show that the




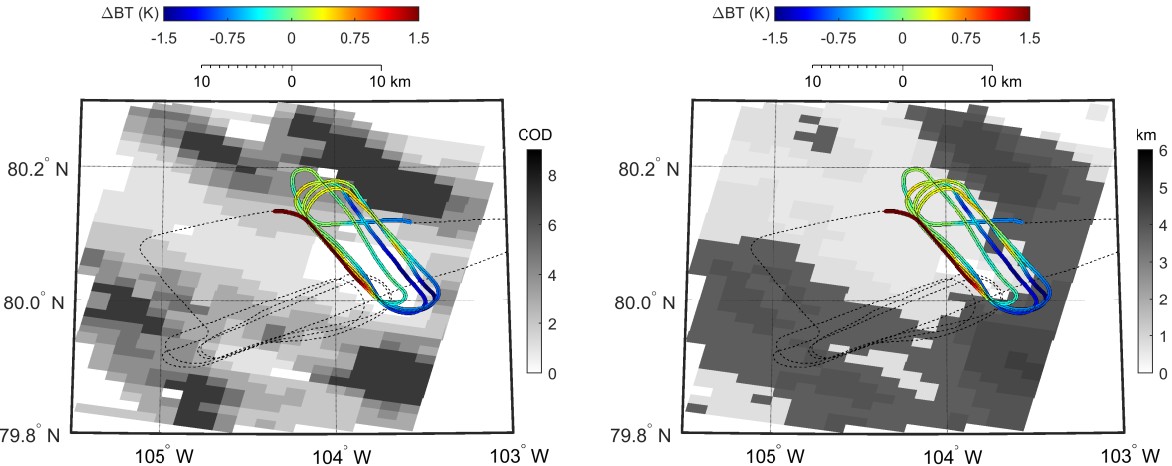

**Figure 10.** (a) Optical depth at 1.24 $\mu$m and (b) cloud top altitude derived from MODIS observations at the beginning of the profile on 13 April (18:09 UTC). Polar 6 trajectory is highlighted, with the color corresponding to the difference between measured and simulated ($\tau = 2$) brightness temperatures for the $10 - 12\ \mu$m channel. Blue means the actual optical depth is less than 2 while red means it is larger.

$30 - 50\ \mu$m channel is mostly sensitive to atmospheric layers below 500 hPa (above $\sim 5$ km), which explains why this channel was not very useful at lower altitude during the campaign. The shorter F-IR wavelengths are sensitive to lower layers of the atmosphere, and window channels are almost insensitive to the atmosphere temperature. These Jacobians also suggest that the 3 channels between 17 and 20 $\mu$m are very similar, making them somehow redundant in such atmospheric conditions. Compar-

ing the absolute values of the Jacobians to the FIRR resolution gives a lower estimate of the vertical resolution the FIRR could reach for profiles retrieval applications. Assuming a resolution of 0.2 K, the corresponding vertical resolution approximately varies from 100 to 200 hPa in F-IR bands. Regarding the FIRR sensitivity to variations in relative humidity, Fig. 11b shows that the $30 - 50\ \mu$m band is the most sensitive, as expected due to the water vapor absorption spectrum. Humidity variations of 5 % for a 100 hPa thick layer above 600 hPa should produce a detectable signal for all F-IR bands, highlighting the potential of the

FIRR for probing humidity profiles in such cold and dry conditions. Note that the Jacobians are positive around the temperature inversion, which is a feature typical of polar conditions. Negative values are consistent with the fact that increasing water vapor increases the greenhouse effect due to the atmosphere and hence decreases radiation at TOA.

To complement this sensitivity analysis, an ice cloud was inserted between 2 and 6 km in the same atmosphere, and the relative humidity with respect to ice correspondingly set to 100 %. Starting from a reference cloud, its optical depth and

effective particle diameter were varied. Figure 12 shows that TOA F-IR brightness temperatures are very sensitive to cloud optical depth, with variations up to 5 K between clear sky conditions and $\tau = 5$. The FIRR resolution approximately converts into a 0.2 resolution in terms of optical depth. The same exercise with varying optical depth shows that for small particles F-IR channels are very sensitive to particle size. However, the sensitivity quickly decreases for largers sizes, which is consistent with the findings of Yang (2003) and Baran (2007), who suggested a sensitivity up to 100 $\mu$m effective dimensions. This sensitivity





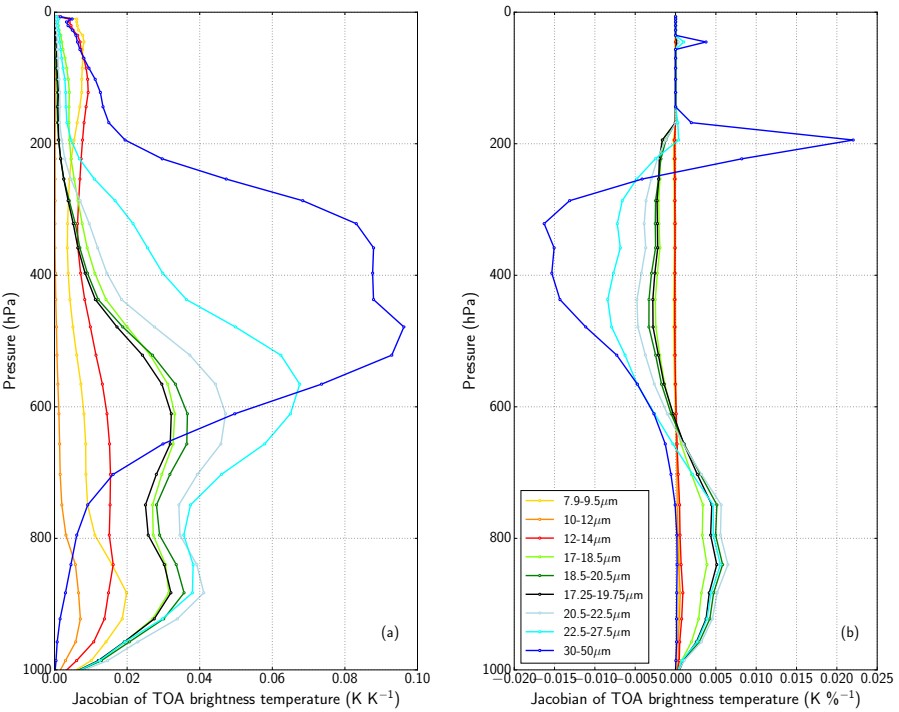

**Figure 11.** (a) Temperature and (b) humidity Jacobians for the TOA brightness temperature for 11 April atmospheric profile. For humidity, variations are in % of the specific humidity.

is directly related to the crystal shape and size distribution assumed for this study, which correspond to cirrus clouds. Although the results above are qualitatively robust, using another ice cloud parameterization could have resulted in different values (e.g. Baran, 2007). In particular, Arctic clouds characterized by rapid crystal growth in high supersaturation conditions may actually have shallower particle size distributions (Jouan et al., 2012) and exhibit more sensitivity to particle size.

## 4.2 Atmospheric cooling rates

Like F-IR radiances, the atmosphere radiative budget is very sensitive to cloud properties. In particular, cloud geometrical and microphysical characteristics largely determine atmospheric cooling rates (Maestri et al., 2005), in such a complex way that clouds can either warm or cool the atmosphere (Maestri, 2003; Lampert et al., 2009). Further understanding of the impact of clouds on the atmosphere radiative budget is of primary importance, and direct measurement of the net radiative fluxes is the best approach to this. The net flux was computed from the broadband sensors, and the cooling rates calculated from its vertical variations (e.g. Cox, 1969). To our knowledge, spectral cooling rates have only been measured once (Mlynczak et al., 2011). During the NETCARE campaign, the FIRR was supposed to have a zenith view to allow net fluxes computation, but shortly before the campaign started this configuration proved to be incompatible in terms of safety. As a consequence, the spectral



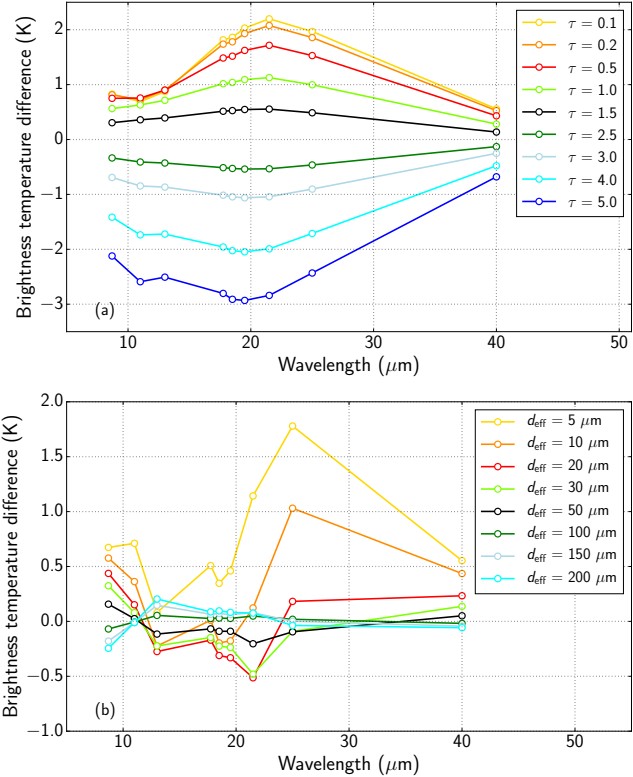

**Figure 12.** TOA brightness temperature differences between various clouds and the reference with $\tau = 2$ ($\tau = 3$ for panel (b)) and $d_{\text{eff}} = 80 \ \mu$m. Panel (a) is for varying optical depth while panel (b) is for varying effective particle diameter.

cooling rates were simulated for 11 April, with clear sky conditions and with a cloud of optical depth 2 and effective particle diameter $80 \ \mu$m. Figure 13a shows that F-IR emission of the atmosphere contributes to cooling the entire atmosphere, resulting in an average 1.5 K day$^{-1}$ cooling in clear sky conditions, which is very consistent with the cooling rates computed from the broadband LW measurements (Fig. 13b). The presence of an ice cloud enhance this cooling since it acts as an efficient radiator

5   (Fig. 13c). The corresponding cooling rates are 2-to-3 times larger than in clear sky conditions (Fig. 13d). Contrary to mid-latitude or tropical conditions, in the absence of solar radiation the cloud radiates more energy than it absorbs from the surface, the latter being too cold to sufficiently warm the cloud. As a consequence, optically thin ice clouds cool the atmosphere in their whole volume, and can dramatically affect the stability of the atmosphere (Blanchet et al., 2011).

### 4.3 Recommendations for future operation

10   The preceding results are now discussed in the framework of planning the TICFIRE satellite mission and in view of future airborne campaigns with the FIRR or similar instruments. First of all, one advantage of using uncooled microbolometers is the possibility to have an imager, as will be the case for TICFIRE. In this study, the FIRR was not used as an imager, though,



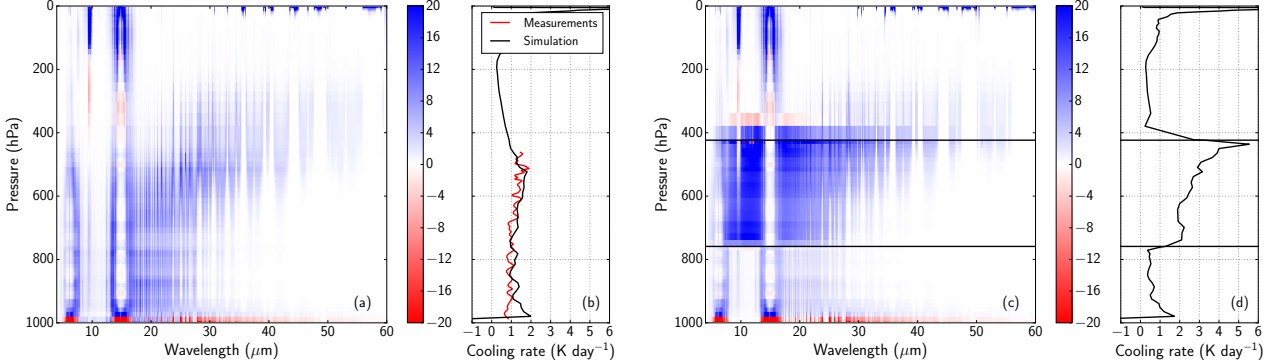

**Figure 13.** Simulated spectral cooling rates for 11 April atmosphere for (a) clear sky and (c) cloudy ($\tau = 2$) conditions. The dark horizontal lines in panel (c) and (d) indicate cloud bottom and top heights. Units is $100 \times$ K day$^{-1}$ $\mu$m$^{-1}$. Simulated broadband cooling rates are also shown (b and d). Broadband cooling rates computed from broadband LW measurements are shown in panel (b).

because it has a much narrower field of fiew than TICFIRE satellite configuration. However it is worth exploring how the accuracy of the measurements would decay if spatial averaging was skipped. To this end, the spectral brightness temperature shown in Fig. 5 is computed again from FIRR measurements, except that spatial averaging is made on 1 (no averaging), 4, 9 or 193 pixels. Nominal data processing is optimized for 193 pixels and could not be applied to a single pixel (Libois et al., 2016),

so that the procedure was slightly changed to ensure that the same calibration is applied independently of the number of pixels averaged. The results are shown in Fig. 14. As expected, spatial averaging improves the repeatability of the measurement, but averaging over 9 pixels already provides a resolution close to 193 pixels. The absolute values are very consistent, with differences less than 0.5 K if more than 1 pixel are used. The remaining differences can be attributed to instrument errors, but scene spatial heterogeneities can not be ruled out. This suggests that the present study is relevant to verify the performances of

the future TICFIRE satellite instrument, whose precision could be increased through spatial averaging over neighbour pixels.

    It is worth pointing out that the NETCARE campaign was not dedicated solely to radiation measurements. Probing ice clouds was one of the objectives, but not the only one. In addition, few clouds were encountered during the campaign and days with too many clouds prevented aircraft operations for safety reasons. Overall the dataset is still modest and further campaigns in the Arctic winter remain necessary. Such campaigns should be dedicated to the radiative properties of ice clouds in order to

maximize the scientific success of this research topic (e.g. CIRCCREX, Fox, 2015). From the FIRR perspective, we noticed that upgrading the current instrument resolution is essential to further constrain radiative transfer simulations and cloud properties retrievals. This can be achieved by improving the environmental conditions of the FIRR within the aircraft, paying more attention to temperature stability. Adding an insulating window to prevent air circulation around the instrument or increasing the pressure inside the instrument to ensure constant outflow from the aircraft would minimize temperature variations. Note

that these recommendations are linked to the fact that Polar 6 cabin is unpressurized and other constraints should be thought of



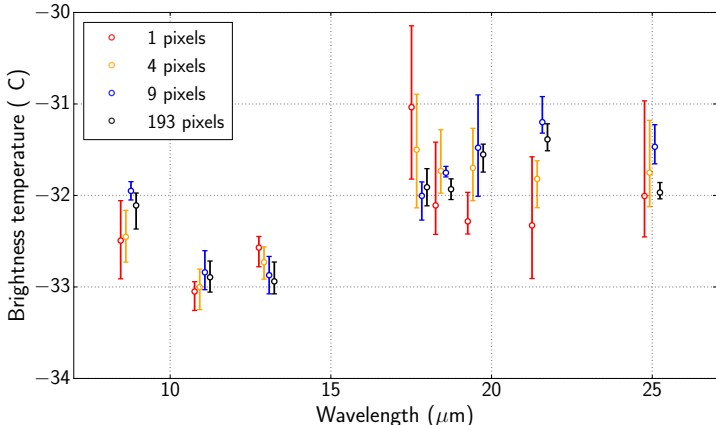

**Figure 14.** FIRR spectral brightness temperatures at 5.56 km as in Fig. 5, except that measurements were averaged over a varying number of pixels, from 1 to 193. Error bars indicate measurement variability along 4 consecutive measurements. For each spectral band the 4 corresponding error bars are slightly displaced horizontally for sake of clarity.

in the case of a pressurized aircraft. Complementary zenith and nadir observations would also be extremely valuable in order to compute cooling rates and sample the whole atmospheric profile.

At the instrument level, the FIRR is the first prototype and improvements are expected from technological developments of uncooled microbolometers, but optimization in the analogical-numerical converter and absence of the detector window in space could already increase the current resolution by a factor of 3 to 5. Likewise, increasing acquisition rate by using a faster filter wheel and scene selection motor would reduce the acquisition time of a sequence by one order of magnitude, thus limiting temperature variations in between calibrations. Such developments are already considered and will be mandatory for the satellite version of the instrument which requires acquisition times around 1 s for a complete scene sequence.

# 5   Conclusions

The first airborne campaign of the FIRR took place in the Arctic and was a great opportunity to study the far-infrared properties of the early spring Arctic atmosphere. Vertical profiles of brightness temperature acquired in clear sky and cloudy conditions provided a strong observational constraint on the radiative properties. At the same time, they increased the limited amount of observations available in the far-infrared, especially in such remote regions. These observations also provided valuable knowledge about the FIRR instrument, which can be used to improve operation and development in view of the TICFIRE satellite mission. This campaign showed that the current state-of-the-art radiative transfer models are well suited for the Arctic and confirm that instrument resolution is better than the uncertainties inherent to the radiative transfer formulation and input observations. They also show that aerosols can significantly impact the radiative budget of the atmosphere, thus implying that a detailed characterization of the aerosols and haze is necessary to refine radiative closure experiments. Although the FIRR





behaved very well during the campaign with respect to its nominal performances, the latter could be improved for accurate retrievals of atmospheric and cloud characteristics. The campaign proved that ice clouds in the Arctic are hard to probe, as much for reasons of safety as for their complexity and their high heterogeneity. Such campaigns should be replicated to improve our understanding of ice cloud formation and radiative properties in polar regions. Accordingly, they should be dedicated to

radiation and combine cloud microphysical observations with various radiation sensors. Such studies are necessary to continue improving our knowledge of ice cloud formation and its parameterization in numerical weather prediction and climate models.

**Data availability**

All NETCARE data will be made public after the end of the project (http://www.netcare-project.ca). In the meantime, access can be granted by contacting the project manager Bob Christensen (bob.christensen@utoronto.ca). The FIRR data used in this

study are available upon request from the authors (libois.quentin@uqam.ca). Requests for access to AWI data should be sent to Martin Gehrmann (martin.gehrmann@awi.de). The CALIPSO and DARDAR products were obtained from the ICARE Data Center (http://www.icare.univ-lille1.fr/). MODIS data were obtained from LAADS (https://ladsweb.nascom.nasa.gov/).

*Author contributions.* Q. Libois, L. Ivanescu and H. Schulz operated the FIRR during the airborne campaign. H. Bozem and W. R. Leaitch were scientific leaders aboard Polar 6 and responsible for gas measurements and cloud probes, respectively. J. Burkart and A. A. Aliabadi

processed the meteorological and aircraft data. Q. Libois and L. Ivanescu processed the data and Q. Libois performed the radiative transfer simulations. Q. Libois wrote the manuscript with contributions of all co-authors.

*Acknowledgements.* This research was funded jointly by the Canadian Space Agency (CSA) through the FAST program, by NETCARE through the Climate Change and Atmospheric Research (CCAR) program at the Natural Sciences and Engineering Research Council of Canada (NSERC), by the Alfred Wegener Institute (AWI) and by Environment and Climate Change Canada (ECCC). We thank the Nunavut

Research Institute and the Nunavut Impact Review Board for licensing the study. We thank Kenn Borek Air Ltd (KBAL), in particular Gary Murtsell and Neil Travers for their skillful piloting across the Arctic. We acknowledge Martin Gehrmann, Manuel Sellmann and Lukas Kandora (AWI) for their technical help during airborne operation. We thank Alexei Korolev (ECCC) for providing Nevzorov and 2D-C probes data and Gerit Birnbaum (AWI) for helping with the processing of AWI radiation sensors. Norm O'Neill (Université de Sherbrooke) accommodated the involvement of L. Ivanescu to the campaign. Canadian Network for the Detection of Atmospheric Change (CANDAC),

to which L. Ivanescu is affiliated, provided logistic support during the stay at Eureka Weather Station. The Institut National d'Optique (INO) provided technical support for the FIRR before and during the campaign. We are grateful to Marcus Dejmek and Daniel Gratton (CSA) for providing logistic resources for the FIRR. We are indebted to Mike Harwood (ECCC) for his direction of the instrument integration and support in the field. We also thank John Ford, David Heath and the University of Toronto machine shop, Jim Hodgson and Lake Central Air Services (LCAS) in Muskoka, Jim Watson (Scale Modelbuilders, Inc.), Andrew Elford (ECCC) and Julia Binder (AWI) for their support of

the instrument integration. We are grateful to Bob Christensen (U. Toronto), Doug MacKenzie (KBAL), Rosa Wu, Carrie Taylor, Sangeeta Sharma, Desiree Toom, Dan Veber, Alina Chivulescu, Andrew Platt, Ralf Staebler, Anne Marie Macdonald and Maurice Watt (ECCC) for their support of the study. We acknowledge modelling support from MPIC and University of Mainz, Mainz (Daniel Kunkel, Jens Krause and



Franziska Köllner) and U. of Quebec at Montreal (Ana Cirisan and a class of students) used for flight planning. We thank the Biogeochemistry department of MPIC for providing the CO instrument and Dieter Scharffe for his support during the preparation phase of the campaign.



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
