# Peer review of "Airborne observations of far-infrared upwelling radiance in the Arctic"

_Atmospheric Chemistry and Physics, 2016_

## Referee Comment (RC1) · Anonymous Referee #2 · 13 Oct 2016

This paper describes the first field application of a far IR radiometer operated on-board the Polar 6 aircraft over Arctic regions during the NETCARE campaign. The paper shows the importance to measure the far IR spectral region and how much these measurements, acquired in all-sky conditions, can improve the sensitivity to specific humidity and the cooling rate of thin ice clouds.

General comment: The paper is well written, clear in the description of the field campaign, and convincing in showing the importance to cover this observational gap in the spectral range of the IR emission. The data analysis is limited to few cases with few general implications for atmospheric science. However, considering that it belongs to the NETCARE special issue, I think that the paper is worth to be published in ACP and of general interest for the Earth radiation budget community.

[Figure]

Some changes are required to improve the figures and the description as indicated here below.

- Introduction. Some more references about the available measurements in Arctic should be added, e.g from ICECAPS experiment or the CANDAC network, in order to better stress the contribution of these new measurements.

- page 5 line 1. Does the same radiometric resolution apply to all the spectral bands ? If not, I would put the numbers in Tab.1 otherwise please clarify the text. Furthermore, is the radiometric resolution limited by the detector noise or by other reasons ? I would add some more information about the noise on the different channels and the related radiometric resolution, even if this is characterized in laboratory conditions.

- page 6 line 2-4. It would be interesting to describe with more details the refinement introduced to better account for quick temperature variations. Otherwise this sentence is too general and not useful.

- page 6 line 17-18. It is not clear whether the images were used or not. If not I would avoid to cite this probe.

- page 7 line 2. 5 cases are too few cases to provide a real overview of the Arctic conditions, they are an example of different conditions. Please rephrase the sentence.

- page 8 sect. 3. Since this paper is published in ACP, even if it is mainly an instrumental paper, I would try to introduce since here the general scientific results expected in the framework of the NETCARE campaign in order to give more evidence to the peculiar results of this work within the general scientific problem of the special issue.

- page 8 line 18. Please clarify whether the value of 0.015 W m-2 sr-1 applies to all the bands.

- page 8 line 25. This sentence is not completely clear because the calibration is not described. Furthermore, Sect 3.1 addresses the radiometric performance in terms of temperature resolution. It would be also interesting to have an idea of the absolute error of the measurement.

- page 11 line 6. I would say a "close agreement" above 2 km, below the difference is always more than 0.6 W m-2.

- page 12 line 3-4. If the peak is not present on the way down, please show this case in the figure.

- page 11 fig. 4. In panel (b) the x-axis label should be Brightness temperature. I would also remove the temperature curve which is also shown in panel (a). Same for Fig. 6 panel (b) and (d).

- page 13 line 10. Do you have some information about these clouds from CALIPSO ?

- page 13 line 17. In the comparison with simulation you should estimate the noise on measurements due to scene variations. Besides the aircraft movement considered here, please add some more considerations at least about the roll of the platform.

- page 13 line 26-27. The sentence "They are of little interest ..." is too general. This spectral range can be of great interest for satellite observations because you can see high altitude clouds.

- page 16 fig.8. As said before, I would use Brightness temperature for x-axis in panel (b) and show the temperature profile only in panel (a). Same for Fig.9.

- page 19 line 6. Since resolution was used for the radiometric measurement, the sentence is not clear. I would say: ... temperature variations of 0.2 K are detectable with a vertical resolution of ...

- General comment on figures. The font size of labels and scales in most of the figures should be enlarged to be clearer.
* * *

---

## Referee Comment (RC2) · Anonymous Referee #1 · 19 Oct 2016

**Airborne observations of far-infrared upwelling radiance in the Arctic**

Quentin Libois et. Al.

**Review of above manuscript:**

The main aim of this paper is to evaluate the performance of the FIRR instrument under field campaign conditions. This is done successfully with overall performance shown to agree with laboratory performance within limitations imposed by the operational and environment conditions. Improving our understanding of the distribution and radiative effects of cirrus clouds in Arctic climates is highly important and TICFIRE a very worthwhile endeavour. Testing and improving the underlying technology for TICFIRE through the FIRR instrument is therefore crucial and this paper highly relevant.

Of the four main objectives mentioned at the end of the introduction I would suggest that the measurements described are not strictly a radiative closure experiment, the atmospheric state is not sufficiently well known to allow this. Similarly for the verification of the spectral signatures of cloud radiance. The work does assess the FIRR radiometric performance and demonstrates the sensitivity of FIRR measurements to atmospheric characteristics.

The inclusion of the section on atmospheric cooling rates is not helpful for the objectives of the paper a fact emphasised by the lack of zenith view data. This can be omitted without impact on the paper.

I would like more detailed information on the in-flight variability of the ancillary data set, such as local humidity and ambient temperature, particularly at fixed flight levels. Please see additional text below.

**Suggested changes to text:**

Replace F-IR with FIR throughout text

Page2

line4: " includes the **strongly absorbing** pure rotation band of water vapor" and coincides with a maximum in the  water vapour continuum strength.

Line 8: "The **emission** maximum of Planck's function…"

Line 11: Reference to the Mars climate sounder is not relevant.

Line 32-33: This is a little confused, the wording may be clearer. "uttermost in Arctic regions because as discussed proportionately more energy is emitted from these

colder surfaces at FIR wavelengths while the same time lower water vapour column increases atmospheric transmission.

Line 22: vignetting by the chimney edges? I assume

Line 28: "One **spectral** measurement thus corresponds to a 0.8 s…"

Page 4:

There needs to be specific reference to the fact that the measurements are comprised from the average of all pixels in the 15 pixel diameter area illuminated by the scene footprint.

The authors highlight the advantages of fast scanning and the high radiometric accuracy of their instrument but in the operational configuration described individual spectral band measurements are, if I understand the text correctly, off-set temporally and hence spatially. This should be made clear at this stage and placed in context to the along track averaging.

The sequence described indicates 0.8 s averaging per band, 9 bands per filter wheel rotation totalling 7.2 s observation time for all bands. Given 3 scene views and 2 calibration scans per cycle that equates to 36 s. The Authors indicate that one complete sequence last 210 s, there is therefore some considerable time unaccounted for, can the Authors expand on this and explain the implications, if any, for high variability scenes such as that observed in the cirrus observations.

Page 5:

Figure 1 does not add a lot to the text and can be omitted
Table 1 would be more informative replaced with a spectral plot showing the filter transmission, similar to that of figure 2a in the Author's earlier paper, "A microbolometer-based far infrared radiometer to study thin ice clouds in the arctic"

Page 6/7:

The description of the flight paths for the aircraft lacks detail, the longitudes indicated on figure 2 (left panel) are wrong (75/60/45 degrees being 15 degrees out). Choose one flight and expand to show detail of the profile track more clearly.

Alternatively a more detailed figure of the flight path could be included with the case details.

Page 8:

Line 6: Is the KT19 spectral response known and has this been applied derive surface temperature with the assumption of a spectrally flat surface emissivity of 0.995, be more explicit.

Line 25-30:

How was the trend in ice temperature over the 30 minutes established, was this correlated against the KT19 data set for validation or was the KT19 data used to establish the trend?

Page 9:

Line3: "To further investigate the reduced **thermal** resolution observed…"

Line 21/22:

"the KT19 was -32.6C **while** a maximum of -24 C **was observed in the atmospheric temperature profile between** 1 and 2 km…"

Line 23: I do not believe you can justify suggesting no cloud above the aircraft from CALIPSO measurements made 3 hours previously, are there MODIS cloud cover products that are nearer in time that you can use.

Line 25: A plot of the atmospheric transmittance vs altitude for each channel may help interpretation.

Page 11:

Figure 4: 4c should indicate how the irradiance measurements were obtained.

Page 12:

Line 1: Be more specific about what feature you are referring to.

Fig 5. Can the Authors include error bars on the simulations using realistic uncertainties applied to the atmospheric data set used in the radiative transfer model.

Page 13:

Lines 19-34: It would be informative to see the spectrally resolved MODTRAN radiance output plotted as brightness temperature with the filter responses superposed, for the 11[th], 20[th] and 21[st] April at the maximum aircraft altitude. Again uncertainties on the simulation BT's would be informative for figure 6.

Page 15:

Figure 7 shows a 2-D image footprint for a 0.8 s scan, can the Authors include the relative positions for all 9 band observations along track for a single filter wheel rotation and indicate the position offsets between filter wheel cycles

Line 3: "This question is le**f**t to future work…"

Line 5-6: You have no uncertainties placed on the MODTRAN simulations so stating the deficiencies here is not justified, for instance what is the along track variation in the measured humidity.

Page 16:

Line 6: ",  with relatively large particles seen **consistently** by the 2D-c probe"

Page 18:

Line 15:

Inferences made from reference to figure 10 would be enhanced with inclusion of a linear plot of relevant data sets as a function of aircraft altitude vs time (location). Co-located MODIS cloud optical depth/height can be superposed for reference.

Page 19:

Line 4: "making them  somewhat redundant….."

Page 20.

Atmospheric cooling rates:

Mlynczak et al 2011, The INFLAME design is such that the net flux is measured directly thus allowing instantaneous cooling rates to be established. It is my understanding that FIRR would require combinations of sequential measurements of zenith and nadir views, similar spectrally resolved measurements of atmospheric cooling rates in the far-infrared have in fact been measured, Harries 2008.

Line 10: "The net flux was computed from broadband sensors". What sensors are these?

The inclusion of this section on cooling rates does not benefit the overall interpretation of the FIRR instrument performance. In itself it is not new nor does it expand on existing work. The "measured" broadband cooling rates are not detailed and the lack of FIRR zenith data is a hindrance.

In my opinion this section should be omitted entirely.

Page 22:

Line 1: "field of  view…."

Line 16: "instrument resolution" What aspect of instrument resolution are you referring to, spectral, spatial, thermal.

---

## Author Comment (AC1) · 29 Nov 2016

**Response to reviewer #2**

The reviewer's comments are in black and our answers are in red.
Modifications of the manuscript are reported in bold and italic.
The pages and lines reported here correspond to the original pdf.

This paper describes the first field application of a far IR radiometer operated on-board the Polar 6 aircraft over Arctic regions during the NETCARE campaign. The paper shows the importance to measure the far IR spectral region and how much these measurements, acquired in all-sky conditions, can improve the sensitivity to specific humidity and the cooling rate of thin ice clouds.

General comment:
The paper is well written, clear in the description of the field campaign, and convincing in showing the importance to cover this observational gap in the spectral range of the IR emission. The data analysis is limited to few cases with few general implications for atmospheric science. However, considering that it belongs to the NETCARE special issue, I think that the paper is worth to be published in ACP and of general interest for the Earth radiation budget community.

Some changes are required to improve the figures and the description as indicated here below.

• Introduction. Some more references about the available measurements in Arctic should be added, e.g from ICECAPS experiment or the CANDAC network, in order to better stress the contribution of these new measurements.

The projects ICECAPS (at Summit) and CANDAC (at Eureka) are now mentioned, to insist that NETCARE contribution is mostly in terms of airborne measurements.

p3 l.4 : "***These scientific flights offered the possibility to probe the atmosphere in situ, thus providing a valuable complement to the extensive ground observations performed at well instrumented sites such as Summit (e.g. ICECAPS project, Shupe el al., 2013) and Eureka (e.g. CANDAC network, Mariani et al., 2012). Altogether, these initiatives aim at refining our understanding of the radiative budget of the Arctic and the critical role clouds play in it, in the continuity of the seminal Surface Heat Budget of the Arctic Ocean (SHEBA) program (e.g. Shupe et al., 2006)***."

• page 5 line 1. Does the same radiometric resolution apply to all the spectral bands ? If not, I would put the numbers in Tab.1 otherwise please clarify the text. Furthermore, is the radiometric resolution limited by the detector noise or by other reasons ? I would add some more information about the noise on the different channels and the related radiometric resolution, even if this is characterized in laboratory conditions.

The Table 1 has been removed and was replaced by a figure showing the spectral transmittances of each channel. The radiometric resolution is very similar from one band to another, because the absorptivity of the detector is spectrally flat (due to the gold black coating) and all filters have maximum transmittance around 80%. The resolution is a bit less, though, for the 22.5-27.5 and 30-50 channels which have slightly lower transmittances of the filter and of the package window, respectively. This is now detailed in the text. Regarding the resolution, in laboratory it is essentially limited by the detector noise.

p5 l.1 : "In this configuration, the radiometric resolution of the FIRR in laboratory conditions ***is***

*essentially limited by detector noise and* is about 0.015 W m$^{-2}$ sr$^{-1}$*. This corresponds to noise equivalent temperature differences of 0.1 – 0.35 K for the range of temperatures investigated in this study. The resolution is nearly constant for the 7 bands ranging from 7.9 to 22.5 µm because the absorptivity of the gold black coating is spectrally uniform and the filters all have similar maximum transmittances. It is approximately 30% less for the filters 22.5 – 27.5 µm and 30 – 50 µm, because of limited filter transmittance for the band 22.5 – 27.5 µm and reduced package window transmittance for the band 30 - 50µm.*"

• page 6 line 2-4. It would be interesting to describe with more details the refinement introduced to better account for quick temperature variations. Otherwise this sentence is too general and not useful.

In Libois et al. (2016) the background radiance is assumed linear in time, and the rate is deduced using three measurements (ABB, HBB, next ABB). Here, another equation is added to the system, namely the next HBB measurement, so that we have 4 equations to retrieve 3 variables instead of 3 equations (eqs. 7 of Libois et al., 2016). Since it is a very technical detail and since the explanation would need too much reference to Libois et al. (2016), we decided to remove this detail.

p6 l.2 : "For previous flights, the calibration procedure detailed in Libois et al. (2016), that takes advantage of non illuminated pixels of the detector to remove the background signal, ensured good quality data for all bands except the 30 – 50 $μm$."

• page 6 line 17-18. It is not clear whether the images were used or not. If not I would avoid to cite this probe.

The probe indicated the presence of large particles, which is used in the analysis, but the exact shape and size were not used because they were not reliable. It has been clarified.

P6 l.17 : "A PMS 2D-C imaging probe *was supposed* to detect larger particles, *but* the images were obscured due to a problem with the true air speed used in the image re-construction, preventing accurate retrieval of particle size distribution. *Practically, this sensor was mostly used to assess the presence of large cloud particles, but did not provide quantitative information about particle shape or size.*"

• page 7 line 2. 5 cases are too few cases to provide a real overview of the Arctic conditions, they are an example of different conditions. Please rephrase the sentence.

"Overview" was replaced by "*samples*"

• page 8 sect. 3. Since this paper is published in ACP, even if it is mainly an instrumental paper, I would try to introduce since here the general scientific results expected in the framework of the NETCARE campaign in order to give more evidence to the peculiar results of this work within the general scientific problem of the special issue.

To present our results in the more general context of the NETCARE campaign, the objectives of the campaign are now presented in more details in Section 2.1. The general context was also recalled in the conclusion. However, we do not dwell too much on the original objectives, because due to the deficiencies in the cloud probe and to the lack of cloud cases, it is hard to derive from this campaign general conclusions regarding the physics of ice clouds in the Arctic.

p4 l.7 : *"One of the objectives was to characterize at the same time the microphysical and the radiative properties of ice clouds, along with the nature of the aerosols, in order to further explore the conditions in which optically thin ice clouds form and how their microphysics depend on background aerosols..*"

p23 l.10 : "The first airborne campaign of the FIRR took place in the Arctic **in the framework of the NETCARE aircraft campaign**. It was a great opportunity to study the **radiative** properties of the early spring Arctic atmosphere, **and highlighted the importance of water vapor and ice clouds in this remote environment**."

• page 8 line 18. Please clarify whether the value of 0.015 W m-2 sr-1 applies to all the bands.

See above.

• page 8 line 25. This sentence is not completely clear because the calibration is not described. Furthermore, Sect 3.1 addresses the radiometric performance in terms of temperature resolution. It would be also interesting to have an idea of the absolute error of the measurement.

This sentenced has been removed because it was confusing. At the same time the description of the BB in Section 2.2.1 has been further detailed. The absolute error is about 0.02 W m$^{-2}$ sr$^{-1}$ according to laboratory experiments.

p4 l.1.25 : "*These correspond to BB nominal temperatures in flight but some experiments were performed with different BB temperatures depending on the environmental constraints, , which is not problematic since the instrument's response is linear in this range of temperature*."

p8 l.17 : "The FIRR performances were investigated based on laboratory and ground-based experiments by Libois et al. (2016). *They estimated a radiometric resolution around 0.015 W m-2 sr-1 and an absolute error of 0.02 W m-2 sr-1, again slightly dependent on the channel considered*.

• page 11 line 6. I would say a "close agreement" above 2 km, below the difference is always more than 0.6 W m-2.

Done. 0.6 W m-2 is now 0.35 W m-2.

• page 12 line 3-4. If the peak is not present on the way down, please show this case in the figure.

This has been added to the figure. Since the descent shows a peak in the opposite direction, it has been mentioned in the manuscript and strenghtens the temperature adjustment hypothesis.

p12 l.3 : "This hypothesis is supported by the fact that data taken on the way down just before starting the ascent **show a peak in the opposite direction**."

• page 11 fig. 4. In panel (b) the x-axis label should be Brightness temperature. I would also remove the temperature curve which is also shown in panel (a). Same for Fig. 6 panel (b) and (d).

Done, as well as for other figures showing vertical profiles of brightness temperature.

• page 13 line 10. Do you have some information about these clouds from CALIPSO ?

CALIPSO does not show any cloud above the aircraft altitude.

• page 13 line 17. In the comparison with simulation you should estimate the noise on measurements due to scene variations. Besides the aircraft movement considered here, please add some more considerations at least about the roll of the platform.

Scene variations do not result in an easily identifiable constant noise. Instead, it is mostly visible when strong variations occur, such as peaks seen on some vertical profiles. The roll of the platform is already meantione p11 l.2, but it is now converted in terms of distance.

p13 l.16 : "a single measurement of 0.8 s spanned 60 m at the surface. *Similarly, a typical roll of 10° during the spiral corresponds to 1 km deviation at the surface when flying at 6 km.* This could generate noise if the surface was not homogeneous at this scale, which was the case at the interface between the sea ice and open water."

• page 13 line 26-27. The sentence "They are of little interest ..." is too general. This spectral range can be of great interest for satellite observations because you can see high altitude clouds.

This point has been detailed.

p13 l.25 : For this reason, the data in the 30 – 50 μm band are not reliable *and are not shown in the rest of the paper. This is not critical in this study because at the flying altitude this band essentially probes local temperature. On the contrary it is expected to be very valuable from a satellite view, where it should provide information about water vapor and clouds at the very top of the troposphere.*"

• page 16 fig.8. As said before, I would use Brightness temperature for x-axis in panel (b) and show the temperature profile only in panel (a). Same for Fig.9.

Done.

• page 19 line 6. Since resolution was used for the radiometric measurement, the sentence is not clear. I would say: ... temperature variations of 0.2 K are detectable with a vertical resolution of ...

p19 l.6 : "*Given the radiometric resolution of the FIRR is about 0.2 K, temperature variations of 0.2 K are detectable with a vertical resolution of 100 to 200 hPa in FIR bands.*"

• General comment on figures. The font size of labels and scales in most of the figures should be enlarged to be clearer.

Done for all concerned figures

**New references**:

Shupe, M. D., Matrosov, S. Y., & Uttal, T. (2006). Arctic mixed-phase cloud properties derived from surface-based sensors at SHEBA. *Journal of the atmospheric sciences*, *63*(2), 697-711.

Shupe, M. D., Turner, D. D., Walden, V. P., Bennartz, R., Cadeddu, M. P., Castellani, B. B., ... & Neely

III, R. R. (2013). High and dry: New observations of tropospheric and cloud properties above the Greenland Ice Sheet. *Bulletin of the American Meteorological Society, 94*(2), 169-186.

---

## Author Comment (AC2) · 29 Nov 2016

**Response to reviewer #1**

The reviewer's comments are in black and our answers are in red.
Modifications of the manuscript are reported in bold and italic.
The pages and lines reported here correspond to the original pdf.

The main aim of this paper is to evaluate the performance of the FIRR instrument under field campaign conditions. This is done successfully with overall performance shown to agree with laboratory performance within limitations imposed by the operational and environment conditions. Improving our understanding of the distribution and radiative effects of cirrus clouds in Arctic climates is highly important and TICFIRE a very worthwhile endeavour. Testing and improving the underlying technology for TICFIRE through the FIRR instrument is therefore crucial and this paper highly relevant.

Of the four main objectives mentioned at the end of the introduction I would suggest that the measurements described are not strictly a radiative closure experiment, the atmospheric state is not sufficiently well known to allow this. Similarly for the verification of the spectral signatures of cloud radiance. The work does assess the FIRR radiometric performance and demonstrates the sensitivity of FIRR measurements to atmospheric characteristics. The inclusion of the section on atmospheric cooling rates is not helpful for the objectives of the paper a fact emphasised by the lack of zenith view data. This can be omitted without impact on the paper. I would like more detailed information on the in-flight variability of the stabilitary data set, such as local humidity and ambient temperature, particularly at fixed flight levels. Please see additional text below.

We believe that the temperature and humidity measurements are sufficient to perform the radiative closure experiment in clear-sky conditions, given that many similar studies have called "radiative closure experiments" comparisons of radiance measurements to simulations fed by radiosoundings data, which is what is presented here. On the contrary, we fully agree that in cloudy conditions we do not have the necessary information to close the radiative experiment. We lack substantial information about cloud properties and the encountered clouds were too heterogeneous. This was already highlighted but is now stated more clearly.

We changed the introduction so that there is less confusion possible between the objectives of the campaign (which include radiative closure in all sky conditions) and those actually achieved. We also explicitly say that the radiative closure is completed for clear-sky conditions, while it is not for cloudy conditions.

p3 l.15 : "In the context of TICFIRE, ***there were four main reasons of*** flying the FIRR in the Arctic: "

p12 l.7 : "FIR simulations provide strong validation of the radiative transfer model, ***resulting in a satisfactory radiative closure for clear-sky conditions.***"

p22 l.13 : "further campaigns in the Arctic winter remain necessary, ***in particular to complete a radiative closure in cloudy conditions, which was not possible here due to lack of quantitative information about clouds properties.***"

p24 l.3 : "and their high heterogeineity. ***As a consequence, measured ice clouds spectral signature could not be compared to simulations with sufficiently well-constrained cloud properties. Such airborne*** campaigns"

As recommended by the reviewer, we removed the section of the discussion dedicated to the cooling rates, because it is mostly based on simulations contrary to the other results. Part of this section has been moved to the introduction to broaden the context of far-infrared radiation in the atmosphere and introduce the notion of efficient atmospheric cooling through LW emission of ice clouds.

Regarding the inflight variability of ancillary data, this is discussed in more details below.

Suggested changes to text:

Replace F-IR with FIR throughout text

done

Page2
line4: "host includes the strongly absorbing pure rotation band of water vapor" and coincides with a maximum in the water vapour continuum strength.

done

Line 8: "The emission maximum of Planck's function..."

done

Line 11: Reference to the Mars climate sounder is not relevant.

This reference has been removed from the introduction to be mentioned only in the presentation of the FIRR instrument, because of the similitude between both FIR filter radiometers. The same is true for the Diviner Lunar Radiometer.

P4 l.17: *In this sense it is very similar to the Mars Climate Sounder (McCleese et al., 2007) and the Diviner Lunar Radiometer Experiment (Paige et al., 2010)*"

Line 32-33: This is a little confused, the wording may be clearer. "uttermost in Arctic regions because as discussed proportionately more energy is emitted from these colder surfaces at FIR wavelengths while the same time lower water vapour column increases atmospheric transmission.

done, with slight modifications.

Line 22: vignetting by the chimney edges? I assume

Actually the edges of the chimney are not in the field of view. The vignetting simply corresponds to standard vignetting, that is the fact pixels on the edge of the illuminated area receive a bit less signal than those in the center.

"to avoid the *small vignetting on the edges of the illuminated area.*"

Line 28: "One spectral measurement thus corresponds to a 0.8 s..."

done

Page 4:

There needs to be specific reference to the fact that the measurements are comprised from the average of all pixels in the 15 pixel diameter area illuminated by the scene footprint. The authors highlight the advantages of fast scanning and the high radiometric accuracy of their instrument but in the operational configuration described individual spectral band measurements are, if I understand the text correctly, off-set temporally and hence spatially. This should be made clear at this stage and placed in context to the along track averaging. The sequence described indicates 0.8 s averaging per band, 9 bands per filter wheel rotation totalling 7.2 s observation time for all bands. Given 3 scene views and 2 calibration scans per cycle that equates to 36 s. The Authors indicate that one complete sequence last 210 s, there is therefore some considerable time unaccounted for, can the Authors expand on this and explain the implications, if any, for high variability scenes such as that observed in the cirrus observations.

To insist on the fact that measurments correspond to spatial averages over the whole illuminated area, we slightly modified the text that was already quite explicit about this:

"In this study, the FIRR is not used as an imager, hence ***the data presented here correspond to averages over*** the selected area of 193 pixels".

This is true, the acquisition of all channels is not simultaneous, hence consecutive measurements do not exactly correspond to the same scene. This is clearly stated now.

"for higher signal levels. ***Note, though, that measurements in successive spectral bands are offset temporally, hence spatially, which has to be borne in mind at the stage of data interpretation, especially when significant scene variations occur in less than 20 s.***"

As for the total duration, it can be roughly decomposed as follows:
- 0.8 * 10 * 5 = 40 s taking measurements on 10 filter wheel  positions (a blank measurement is also taken)
- 1.5 * 17 * 5 = 126 s rotating the filter wheel (1.5 s to move of one position, total of 17 positions on the filter wheel)
- 3*15 s = 45 s rotating the pointing mirror

This means that a lot of time is lost rotating the filter wheel and the pointing mirror, which is one of the major issues that we should work on in the future.

This is now detailed:

"One FIRR measurement sequence lasts 210 s, ***during which approximately 40 s are used to actually take measurements and 170 s are spent rotating the filter wheel and the pointing mirror***. A sequence consists of [...]".

"that measures all 9 filters ***in approximately 20 s***"

The impact of this temporal offset is already discussed in Fig. 4 that shows apparent spikes in brightness temperatures.

The impact in the case of hight variability scenes is now detailed in the last section of the discussion dedicated to the recommendations for future operations:

p23 l.7 : *It would also ensure that measurements in all channels are taken on the same target, which was not always the case during the campaign above leads or through highly heterogeneous ice clouds.* Such **technical** developments"

Page 5:
Figure 1 does not add a lot to the text and can be omitted Table 1 would be more informative replaced with a spectral plot showing the filter transmission, similar to that of figure 2a in the Author's earlier paper, "A microbolometer-based far infrared radiometer to study thin ice clouds in the arctic".

We removed Figure 1 but moved the picture 1b to the paragraph describing the issue we had with the input of air inside the instrument, the latter being difficult to understand without the support of such a picture of the hatch. As suggested, we replaced Table 1 by the filters transmittance, indicating the band pass in the legend of the figure.

[Figure]

Page 6/7:
The description of the flight paths for the aircraft lacks detail, the longitudes indicated on figure 2 (left panel) are wrong (75/60/45 degrees being 15 degrees out). Choose one flight and expand to show detail of the profile track more clearly. Alternatively a more detailed figure of the flight path could be included with the case details.

The longitudes were updated because they were indeed 15 degrees off.
A detailed flight path for the 11 April flight has been added to Figure 2. It shows the size of the spirals and the trajectory typical for a vertical profile at constant speed. The color indicates the altitude.

[Figure]

Page 8:
Line 6: Is the KT19 spectral response known and has this been applied derive surface temperature with the assumption of a spectrally flat surface emissivity of 0.995, be more explicit.

It was assumed that the KT19 measures the radiation in the range 9.6-11.5 μm (square response) and that in this range surface emissivty is flat at 0.995. This is now detailed.

P8 l.6 : "from the KT19 observations assuming a *uniform spectral response of the instrument and a spectrally flat* surface emissivity of 0.995 *in the range 9.6-11.5 μm*."

Line 25-30:
How was the trend in ice temperature over the 30 minutes established, was this correlated against the KT19 data set for validation or was the KT19 data used to establish the trend?

This experiment was performed on snow when the aircraft was on the ground, so that only the FIRR was operating. The KT19 was not. Here we're interested in the resolution of the measurement, so that we removed the monotonic temporal trend attributed to snow temperature variation. This is now stated more clearly.

P8 l.28 : "for each spectral band. *For all bands, the radiance increased continuously throughout the experiment, which was attributed to an increase of snow temperature. To remove this effect and focus on the resolution of the measurement only, the radiance series were first detrended,* and the standard deviation of the residual was then computed."

Page 9:
Line3: "To further investigate the reduced thermal resolution observed…"

done

Line 21/22:
"the KT19 was -32.6C while a maximum of -24 C was observed in the atmospheric temperature profile between 1 and 2 km…"

done

Line 23: I do not believe you can justify suggesting no cloud above the aircraft from CALIPSO measurements made 3 hours previously, are there MODIS cloud cover products that are nearer in time that you can use.

This is true. An Aqua MODIS image was taken above the flight area at 6:45 PM (see images below), while the spiral ascent took place between 7:00 and 7:55 PM. This picture and the corresponding cloud products show a very large clear sky area around the flight area. The text was changing accordingly.

p7 l.2 : "*Images taken by the MODIS and the associated cloud products are also used to investigate cloud conditions above the aircraft*."

p9 l.23 : "and the *Aqua MODIS image taken at 18:45 UTC shows* that no clouds"

[Figure]

[Figure]

True color image (left) and cloud optical depth(right) from Aqua MODIS at 6:45 PM on 11 April

Line 25: A plot of the atmospheric transmittance vs altitude for each channel may help interpretation.

Following this suggestion the following figure was added. It shows the distance from the aircraft such that the atmospheric transmittance reaches 75%. It gives an idea of the distance to which each channel penetrates, which helps to interpret the radiance profiles shown in Fig. 4.

[Figure]

Some text was added accordingly:

p9 l.25 : "*To further illustrate this differential sensitivity to the temperature profile, Fig. 6 shows the penetration depth of each channel as a function of altitude. The channels that penetrate the least are sensitive to the conditions closest below the aircraft*."

Page 11:
Figure 4: 4c should indicate how the irradiance measurements were obtained.

"Vertical profiles of (a) temperature and relative humidity ***measured by in situ probes***, (b) FIRR brightness temperatures and (c) upwelling broadband LW irradiance ***measured by the CGR-4 pyrgeometer*** for 11 April flight."

Page 12:
Line 1: Be more specific about what feature you are referring to.

We clarified this:

p12 l.1 : "measurements show an unexpected ***peaked minimum***. Although the origin of ***this peak*** is not ***fully*** understood"

Fig 5. Can the Authors include error bars on the simulations using realistic uncertainties applied to the atmospheric data set used in the radiative transfer model.

Complementary simulations were performed for the 11 April flight, namely one with humidity increased by 2.5% and temperature increased by 0.3 K, the other with humidity decreased by 2.5% and temperature decreased by 0.3 K. These uncertainties correspond to the uncertainties of the temperature and humidy measurements. These simulations were used to estimate error bars in Fig. 5 (see below).

[Figure]

The text was also modified as follows:

p12 l.11 : "***In addition, most deviations between observations and simulations are within the range of uncertainties due to uncertainties of the temperature and relative humidity measurements.***"

Page 13:
Lines 19-34: It would be informative to see the spectrally resolved MODTRAN radiance output plotted as brightness temperature with the filter responses superposed, for the 11th, 20th and 21st April at the maximum aircraft altitude. Again uncertainties on the simulation BT's would be informative for figure 6.

The following figure was added to show the simulated high resolution brightness temperatures. The 21 April case is not shown because it is somehow redundant with the 20. The text was updated accordingly.

[Figure]

p13 l. 15 : *"**The difference between the conditions encountered on 11 and 20 April is further illustrated in Fig. 9. It shows the high spectral resolution brightness temperature simulated by MODTRAN at 6 km altitude for both flights, and the corresponding simulated FIRR spectral signatures. This highlights the greater transparency of the atmosphere in the FIR for the 11 April**."*

Page 15:
Figure 7 shows a 2-D image footprint for a 0.8 s scan, can the Authors include the relative positions for all 9 band observations along track for a single filter wheel rotation and indicate the position offsets between filter wheel cycles

We added some circles on the image to indicate when the first (plain line) and last (dashed) filters of a sequence are mesaured.

[Figure]

Line 3: "This question is left to future work..."

Line 5-6: You have no uncertainties placed on the MODTRAN simulations so stating the deficiencies here is not justified, for instance what is the along track variation in the measured humidity.

We show below the high temporal resolution measurements of water vapor, along with the "average" profile used for the MODTRAN simulation of 20 April. No significant variations of the water vapor are observed along track. The figure is not shown in the manuscript but this potential source of error is ruled out.

[Figure]

p13 l.32 : "*In addition, water vapor measurements along track did not show significant variability, so that spatial variability of water vapor can be ruled out. Only the incursion of a wet air mass below the aircraft before the end of the ascent could explain such a discrepancy between observations and simulations. In such case the water vapor profile used in the simulation would not correspond to the actual profile at the time of the measurement, but this is unlikely given that it was observed on two different flights*."

Page 16:
Line 6: ", consistently with relatively large particles seen consistently by the 2D-C probe"

Page 18:
Line 15:
Inferences made from reference to figure 10 would be enhanced with inclusion of a linear plot of relevant data sets as a function of aircraft altitude vs time (location). Colocated MODIS cloud optical depth/height can be superposed for reference.

The Figure 10 has been redrawn because the data shown were erroneously spatially interpolated. The new MODIS maps have been shifted by a few pixels, such that the interpretation is a bit changed in the manuscript.

Also, we added the timeseries of the brightness temperature difference, along with altitude and the time series of cloud optical depth and cloud top height corresponding to the maps shown in Fig. 10.

[Figure]

p18 l.14 : "In fact, the difference between the temperature measured by the 10- 12 µm channel and the simulation with τ = 2 (indicated by the color of the trajectory in Fig. 13) is minimum *near the area corresponding to the high altitude cloud, which suggests that the cloud there has an optical depth larger than 2*. It is higher elsewhere, meaning that FIRR senses warmer temperatures corresponding to either a thinner or lower cloud. *The variations of the brightness temperature difference are more evident in Fig. 13c, that shows the time series of the difference along with the MODIS estimates of cloud characteristics.*"

Page 19:
Line 4: "making them somehow somewhat redundant....."

done

Page 20.
Atmospheric cooling rates:
Mlynczak et al 2011, The INFLAME design is such that the net flux is measured directly thus allowing instantaneous cooling rates to be established. It is my understanding that FIRR would require combinations of sequential measurements of zenith and nadir views, similar spectrally resolved measurements of atmospheric cooling rates in the far-infrared have in fact been measured, Harries 2008.

The section of the discussion dedicated to the cooling rates has been removed. Part of this has been moved to the introduction and recommendations for future operations.

Line 10: "The net flux was computed from broadband sensors". What sensors are these?
The inclusion of this section on cooling rates does not benefit the overall interpretation of the FIRR instrument performance. In itself it is not new nor does it expand on existing work. The "measured" broadband cooling rates are not detailed and the lack of FIRR zenith data is a hindrance. In my opinion this section should be omitted entirely.

The section has been omitted as suggested.

Page 22:
Line 1: "field of fiew view...."

done

Line 16: "instrument resolution" What aspect of instrument resolution are you referring to, spectral, spatial, thermal.

We added "***radiometric***"